# Ultrahyperbolic Representation Learning

**Marc T. Law**  **Jos Stam**

NVIDIA

## Abstract

In machine learning, data is usually represented in a (flat) Euclidean space where distances between points are along straight lines. Researchers have recently considered more exotic (non-Euclidean) Riemannian manifolds such as hyperbolic space which is well suited for tree-like data. In this paper, we propose a representation living on a pseudo-Riemannian manifold of constant nonzero curvature. It is a generalization of hyperbolic and spherical geometries where the nondegenerate metric tensor need not be positive definite. We provide the necessary learning tools in this geometry and extend gradient-based optimization techniques. More specifically, we provide closed-form expressions for distances via geodesics and define a descent direction to minimize some objective function. Our novel framework is applied to graph representations.

## 1   Introduction

In most machine learning applications, data representations lie on a smooth manifold [16] and the training procedure is optimized with an iterative algorithm such as line search or trust region methods [20]. In most cases, the smooth manifold is Riemannian, which means that it is equipped with a positive definite metric. Due to the positive definiteness of the metric, the negative of the (Riemannian) gradient is a descent direction that can be exploited to iteratively minimize some objective function [1].

The choice of metric on the Riemannian manifold determines how relations between points are quantified. The most common Riemannian manifold is the flat Euclidean space, which has constant zero curvature and the distances between points are measured by straight lines. An intuitive example of non-Euclidean Riemannian manifold is the spherical model (*i.e.* representations lie on a sphere) that has constant positive curvature and is used for instance in face recognition [25, 26]. On the sphere, geodesic distances are a function of angles. Similarly, Riemannian spaces of constant negative curvature are called hyperbolic [23]. Such spaces were shown by Gromov to be well suited to represent tree-like structures [10]. The machine learning community has adopted these spaces to learn tree-like graphs [5] and hierarchical data structures [11, 18, 19], and also to compute means in tree-like shapes [6, 7].

In this paper, we consider a class of pseudo-Riemannian manifolds of constant nonzero curvature [28] not previously considered in machine learning. These manifolds not only generalize the hyperbolic and spherical geometries mentioned above, but also contain hyperbolic and spherical submanifolds and can therefore describe relationships specific to those geometries. The difference is that we consider the larger class of pseudo-Riemannian manifolds where the considered nondegenerate metric tensor need not be positive definite. Optimizing a cost function on our non-flat ultrahyperbolic space requires a descent direction method that follows a path along the curved manifold. We achieve this by employing tools from differential geometry such as geodesics and exponential maps. The theoretical contributions in this paper are two-fold: (1) explicit methods to calculate dissimilarities and (2) general optimization tools on pseudo-Riemannian manifolds of constant nonzero curvature.

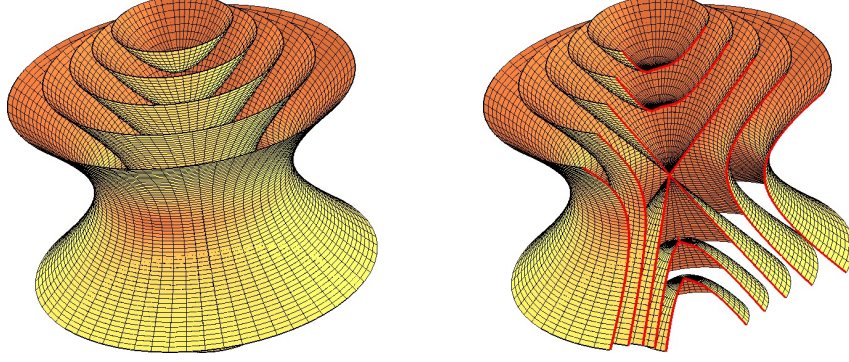

Figure 1: Two iso-surface depictions of an orthographic projection of the same pseudo-hyperboloid $\mathcal{Q}_{-1}^{2,1}$ into $\mathbb{R}^3$ along one time dimension. It contains the entire family of hyperboloids as submanifolds.

## 2 Pseudo-Hyperboloids

**Notation:** We denote points on a smooth manifold $\mathcal{M}$ [16] by boldface Roman characters $\mathbf{x} \in \mathcal{M}$. $T_\mathbf{x}\mathcal{M}$ is the tangent space of $\mathcal{M}$ at $\mathbf{x}$ and we write tangent vectors $\boldsymbol{\xi} \in T_\mathbf{x}\mathcal{M}$ in boldface Greek fonts. $\mathbb{R}^d$ is the (flat) $d$-dimensional Euclidean space, it is equipped with the (positive definite) dot product denoted by $\langle \cdot, \cdot \rangle$ and defined as $\langle \mathbf{x}, \mathbf{y} \rangle = \mathbf{x}^\top \mathbf{y}$. The $\ell_2$-norm of $\mathbf{x}$ is $\|\mathbf{x}\| = \sqrt{\langle \mathbf{x}, \mathbf{x} \rangle}$. $\mathbb{R}_*^d = \mathbb{R}^d \backslash \{\mathbf{0}\}$ is the Euclidean space with the origin removed.

**Pseudo-Riemannian manifolds:** A smooth manifold $\mathcal{M}$ is pseudo-Riemannian (also called semi-Riemannian [21]) if it is equipped with a pseudo-Riemannian metric tensor (named "metric" for short in differential geometry). The pseudo-Riemannian metric $g_\mathbf{x} : T_\mathbf{x}\mathcal{M} \times T_\mathbf{x}\mathcal{M} \to \mathbb{R}$ at some point $\mathbf{x} \in \mathcal{M}$ is a nondegenerate symmetric bilinear form. Nondegeneracy means that if for a given $\boldsymbol{\xi} \in T_\mathbf{x}\mathcal{M}$ and for all $\boldsymbol{\zeta} \in T_\mathbf{x}\mathcal{M}$ we have $g_\mathbf{x}(\boldsymbol{\xi}, \boldsymbol{\zeta}) = 0$, then $\boldsymbol{\xi} = \mathbf{0}$. If the metric is also positive definite (*i.e.* $\forall \boldsymbol{\xi} \in T_\mathbf{x}\mathcal{M}$, $g_\mathbf{x}(\boldsymbol{\xi}, \boldsymbol{\xi}) > 0$ iff $\boldsymbol{\xi} \neq \mathbf{0}$), then it is Riemannian. Riemannian geometry is a special case of pseudo-Riemannian geometry where the metric is positive definite. In general, this is not the case and non-Riemannian manifolds distinguish themselves by having some non-vanishing tangent vectors $\boldsymbol{\xi} \neq \mathbf{0}$ that satisfy $g_\mathbf{x}(\boldsymbol{\xi}, \boldsymbol{\xi}) \leq 0$. We refer the reader to [2, 21, 28] for details.

**Pseudo-hyperboloids** generalize spherical and hyperbolic manifolds to the class of pseudo-Riemannian manifolds. Let us note $d = p + q + 1 \in \mathbb{N}$ the dimensionality of some pseudo-Euclidean space where each vector is written $\mathbf{x} = (x_0, x_1, \cdots, x_{q+p})^\top$. That space is denoted by $\mathbb{R}^{p,q+1}$ when it is equipped with the following scalar product (*i.e.* nondegenerate symmetric bilinear form [21]):

$$\forall \mathbf{a} = (a_0, \cdots, a_{q+p})^\top, \ \mathbf{b} = (b_0, \cdots, b_{q+p})^\top, \ \langle \mathbf{a}, \mathbf{b} \rangle_q = -\sum_{i=0}^{q} a_i b_i + \sum_{j=q+1}^{p+q} a_j b_j = \mathbf{a}^\top \mathbf{G} \mathbf{b}, \ (1)$$

where $\mathbf{G} = \mathbf{G}^{-1} = \mathbf{I}_{q+1,p}$ is the $d \times d$ diagonal matrix with the first $q+1$ diagonal elements equal to $-1$ and the remaining $p$ equal to $1$. Since $\mathbb{R}^{p,q+1}$ is a vector space, we can identify the tangent space to the space itself by means of the natural isomorphism $T_\mathbf{x}\mathbb{R}^{p,q+1} \approx \mathbb{R}^{p,q+1}$. Using the terminology of special relativity, $\mathbb{R}^{p,q+1}$ has $q+1$ time dimensions and $p$ space dimensions.

A pseudo-hyperboloid is the following submanifold of codimension one (*i.e.* hypersurface) in $\mathbb{R}^{p,q+1}$:

$$\mathcal{Q}_\beta^{p,q} = \left\{ \mathbf{x} = (x_0, x_1, \cdots, x_{p+q})^\top \in \mathbb{R}^{p,q+1} : \|\mathbf{x}\|_q^2 = \beta \right\}, \quad (2)$$

where $\beta \in \mathbb{R}_*$ is a nonzero real number and the function $\| \cdot \|_q^2$ given by $\|\mathbf{x}\|_q^2 = \langle \mathbf{x}, \mathbf{x} \rangle_q$ is the associated quadratic form of the scalar product. It is equivalent to work with either $\mathcal{Q}_\beta^{p,q}$ or $\mathcal{Q}_{-\beta}^{q+1,p-1}$ as they are interchangeable via an anti-isometry (see supp. material). For instance, the unit $q$-sphere $\mathcal{S}^q = \left\{ \mathbf{x} \in \mathbb{R}^{q+1} : \|\mathbf{x}\| = 1 \right\}$ is anti-isometric to $\mathcal{Q}_{-1}^{0,q}$ which is then spherical. In the literature, the set $\mathcal{Q}_\beta^{p,q}$ is called a "pseudo-sphere" when $\beta > 0$ and a "pseudo-hyperboloid" when $\beta < 0$. In the rest of the paper, we only consider the pseudo-hyperbolic case (*i.e.* $\beta < 0$). Moreover, for any $\beta < 0$, $\mathcal{Q}_\beta^{p,q}$ is homothetic to $\mathcal{Q}_{-1}^{p,q}$, the value of $\beta$ can then be considered to be $-1$. We can obtain the

spherical and hyperbolic geometries by constraining all the elements of the space dimensions of a pseudo-hyperboloid to be zero or constraining all the elements of the time dimensions except one to be zero, respectively. Pseudo-hyperboloids then generalize spheres and hyperboloids.

The pseudo-hyperboloids that we consider in this paper are hard to visualize as they live in ambient spaces with dimension higher than 3. In Fig. 1, we show iso-surfaces of a projection of the 3-dimensional pseudo-hyperboloid $\mathcal{Q}_{-1}^{2,1}$ (embedded in $\mathbb{R}^{2,2}$) into $\mathbb{R}^3$ along its first time dimension.

**Metric tensor and tangent space:** The metric tensor at $\mathbf{x} \in \mathcal{Q}_{\beta}^{p,q}$ is $g_{\mathbf{x}}(\cdot, \cdot) = \langle \cdot, \cdot \rangle_q$ where $g_{\mathbf{x}} : T_{\mathbf{x}} \mathcal{Q}_{\beta}^{p,q} \times T_{\mathbf{x}} \mathcal{Q}_{\beta}^{p,q} \to \mathbb{R}$. By using the isomorphism $T_{\mathbf{x}} \mathbb{R}^{p,q+1} \approx \mathbb{R}^{p,q+1}$ mentioned above, the tangent space of $\mathcal{Q}_{\beta}^{p,q}$ at $\mathbf{x}$ can be defined as $T_{\mathbf{x}} \mathcal{Q}_{\beta}^{p,q} = \left\{ \boldsymbol{\xi} \in \mathbb{R}^{p,q+1} : \langle \mathbf{x}, \boldsymbol{\xi} \rangle_q = 0 \right\}$ for all $\beta \neq 0$. Finally, the orthogonal projection of an arbitrary $d$-dimensional vector $\mathbf{z}$ onto $T_{\mathbf{x}} \mathcal{Q}_{\beta}^{p,q}$ is:

$$\Pi_{\mathbf{x}}(\mathbf{z}) = \mathbf{z} - \frac{\langle \mathbf{z}, \mathbf{x} \rangle_q}{\langle \mathbf{x}, \mathbf{x} \rangle_q} \mathbf{x}. \tag{3}$$

## 3  Measuring Dissimilarity on Pseudo-Hyperboloids

This section introduces the differential geometry tools necessary to quantify dissimilarities/distances between points on $\mathcal{Q}_{\beta}^{p,q}$. Measuring dissimilarity is an important task in machine learning and has many applications (*e.g.* in metric learning [29]).

**Intrinsic geometry:** The *intrinsic geometry* of the hypersurface $\mathcal{Q}_{\beta}^{p,q}$ embedded in $\mathbb{R}^{p,q+1}$ (*i.e.* the geometry perceived by the inhabitants of $\mathcal{Q}_{\beta}^{p,q}$ [21]) derives solely from its metric tensor applied to tangent vectors to $\mathcal{Q}_{\beta}^{p,q}$. For instance, it can be used to measure the arc length of a tangent vector joining two points along a geodesic and define their geodesic distance. Before considering geodesic distances, we consider extrinsic distances (*i.e.* distances in the ambient space $\mathbb{R}^{p,q+1}$). Since $\mathbb{R}^{p,q+1}$ is isomorphic to its tangent space, tangent vectors to $\mathbb{R}^{p,q+1}$ are naturally identified with points. Using the quadratic form of Eq. (1), the extrinsic distance between two points $\mathbf{a}, \mathbf{b} \in \mathcal{Q}_{\beta}^{p,q}$ is:

$$\mathrm{d}_q(\mathbf{a}, \mathbf{b}) = \sqrt{|\|\mathbf{a} - \mathbf{b}\|_q^2|} = \sqrt{|\|\mathbf{a}\|_q^2 + \|\mathbf{b}\|_q^2 - 2\langle \mathbf{a}, \mathbf{b} \rangle_q|} = \sqrt{|2\beta - 2\langle \mathbf{a}, \mathbf{b} \rangle_q|}. \tag{4}$$

This distance is a good proxy for the geodesic distance $\mathrm{d}_\gamma(\cdot, \cdot)$, that we introduce below, if it preserves *distance relations*: $\mathrm{d}_\gamma(\mathbf{a}, \mathbf{b}) < \mathrm{d}_\gamma(\mathbf{c}, \mathbf{d})$ iff $\mathrm{d}_q(\mathbf{a}, \mathbf{b}) < \mathrm{d}_q(\mathbf{c}, \mathbf{d})$. This relation is satisfied for two special cases of pseudo-hyperboloids for which the geodesic distance is well known:

• **Spherical manifold** ($\mathcal{Q}_{\beta}^{0,q}$)**:** If $p = 0$, the geodesic distance $\mathrm{d}_\gamma(\mathbf{a}, \mathbf{b}) = \sqrt{|\beta|} \cos^{-1}\left( \frac{\langle \mathbf{a}, \mathbf{b} \rangle_q}{\beta} \right)$ is called spherical distance. In practice, the cosine similarity $\frac{\langle \cdot, \cdot \rangle_q}{\beta}$ is often considered instead of $\mathrm{d}_\gamma(\cdot, \cdot)$ since it satisfies $\mathrm{d}_\gamma(\mathbf{a}, \mathbf{b}) < \mathrm{d}_\gamma(\mathbf{c}, \mathbf{d})$ iff $\langle \mathbf{a}, \mathbf{b} \rangle_q < \langle \mathbf{c}, \mathbf{d} \rangle_q$ iff $\mathrm{d}_q(\mathbf{a}, \mathbf{b}) < \mathrm{d}_q(\mathbf{c}, \mathbf{d})$.

• **Hyperbolic manifold (upper sheet of the two-sheet hyperboloid $\mathcal{Q}_{\beta}^{p,0}$):** If $q = 0$, the geodesic distance $\mathrm{d}_\gamma(\mathbf{a}, \mathbf{b}) = \sqrt{|\beta|} \cosh^{-1}\left( \frac{\langle \mathbf{a}, \mathbf{b} \rangle_q}{\beta} \right)$ with $a_0 > 0$ and $b_0 > 0$ is called Poincaré distance [19]. The (extrinsic) Lorentzian distance was shown to be a good proxy in hyperbolic geometry [11].

For the **ultrahyperbolic** case (*i.e.* $q \geq 1$ and $p \geq 2$), the distance relations are not preserved: $\mathrm{d}_\gamma(\mathbf{a}, \mathbf{b}) < \mathrm{d}_\gamma(\mathbf{c}, \mathbf{d}) \iff\!\!\!\!\!/ \ \mathrm{d}_q(\mathbf{a}, \mathbf{b}) < \mathrm{d}_q(\mathbf{c}, \mathbf{d})$. We then need to consider only geodesic distances. This section introduces closed-form expressions for geodesic distances on ultrahyperbolic manifolds.

**Geodesics:** Informally, a geodesic is a curve joining points on a manifold $\mathcal{M}$ that minimizes some "effort" depending on the metric. More precisely, let $I \subseteq \mathbb{R}$ be a (maximal) interval containing 0. A geodesic $\gamma : I \to \mathcal{M}$ maps a real value $t \in I$ to a point on the manifold $\mathcal{M}$. It is a curve on $\mathcal{M}$ defined by its initial point $\gamma(0) = \mathbf{x} \in \mathcal{M}$ and initial tangent vector $\gamma'(0) = \boldsymbol{\xi} \in T_{\mathbf{x}} \mathcal{M}$ where $\gamma'(t)$ is the derivative of $\gamma$ at $t$. By analogy with physics, $t$ is considered as a time value. Intuitively, one can think of the curve as the trajectory over time of a ball being pushed from a point $\mathbf{x}$ at $t = 0$ with initial velocity $\boldsymbol{\xi}$ and constrained to roll on the manifold. We denote this curve explicitly by $\gamma_{\mathbf{x} \to \boldsymbol{\xi}}(t)$ unless the dependence is obvious from the context. For this curve to be a geodesic, its acceleration has to be zero: $\forall t \in I, \gamma''(t) = \mathbf{0}$. This condition is a second-order ordinary differential equation that has a unique solution for a given set of initial conditions [17]. The interval $I$ is said to be maximal if it cannot be extended to a larger interval. In the case of $\mathcal{Q}_{\beta}^{p,q}$, we have $I = \mathbb{R}$ and $I$ is then maximal.

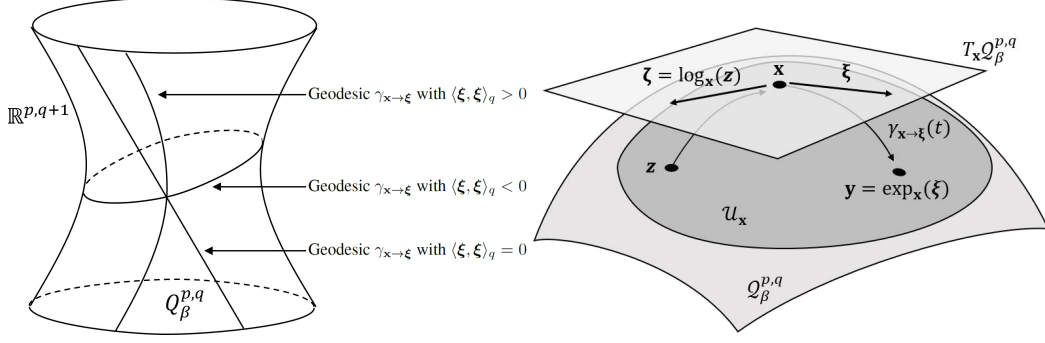

Figure 2: (left) Illustration of the three types of geodesics of a pseudo-hyperboloid defined in Eq. (5). (right) Exponential map and logarithm map.

**Geodesic of $\mathcal{Q}_\beta^{p,q}$:** As we show in the supp. material, the geodesics of $\mathcal{Q}_\beta^{p,q}$ are a combination of the hyperbolic, flat and spherical cases. The nature of the geodesic $\gamma_{\mathbf{x}\to\boldsymbol{\xi}}$ depends on the sign of $\langle\boldsymbol{\xi},\boldsymbol{\xi}\rangle_q$. For all $t\in\mathbb{R}$, the geodesic $\gamma_{\mathbf{x}\to\boldsymbol{\xi}}$ of $\mathcal{Q}_\beta^{p,q}$ with $\beta<0$ is written:

$$\gamma_{\mathbf{x}\to\boldsymbol{\xi}}(t) = \begin{cases} \cosh\left(\frac{t\sqrt{|\langle\boldsymbol{\xi},\boldsymbol{\xi}\rangle_q|}}{\sqrt{|\beta|}}\right)\mathbf{x} + \frac{\sqrt{|\beta|}}{\sqrt{|\langle\boldsymbol{\xi},\boldsymbol{\xi}\rangle_q|}}\sinh\left(\frac{t\sqrt{|\langle\boldsymbol{\xi},\boldsymbol{\xi}\rangle_q|}}{\sqrt{|\beta|}}\right)\boldsymbol{\xi} & \text{if } \langle\boldsymbol{\xi},\boldsymbol{\xi}\rangle_q > 0 \\ \mathbf{x} + t\boldsymbol{\xi} & \text{if } \langle\boldsymbol{\xi},\boldsymbol{\xi}\rangle_q = 0 \\ \cos\left(\frac{t\sqrt{|\langle\boldsymbol{\xi},\boldsymbol{\xi}\rangle_q|}}{\sqrt{|\beta|}}\right)\mathbf{x} + \frac{\sqrt{|\beta|}}{\sqrt{|\langle\boldsymbol{\xi},\boldsymbol{\xi}\rangle_q|}}\sin\left(\frac{t\sqrt{|\langle\boldsymbol{\xi},\boldsymbol{\xi}\rangle_q|}}{\sqrt{|\beta|}}\right)\boldsymbol{\xi} & \text{if } \langle\boldsymbol{\xi},\boldsymbol{\xi}\rangle_q < 0 \end{cases}$$ (5)

We recall that $\langle\boldsymbol{\xi},\boldsymbol{\xi}\rangle_q = 0$ does not imply $\boldsymbol{\xi} = \mathbf{0}$. The geodesics are an essential ingredient to define a mapping known as the exponential map. See Fig. 2 (left) for a depiction of these three types of geodesics, and Fig. 2 (right) for a depiction of the other quantities introduced in this section.

**Exponential map:** Exponential maps are a way of collecting all of the geodesics of a pseudo-Riemannian manifold $\mathcal{M}$ into a unique differentiable mapping. Let $\mathcal{D}_\mathbf{x} \subseteq T_\mathbf{x}\mathcal{M}$ be the set of tangent vectors $\boldsymbol{\xi}$ such that $\gamma_{\mathbf{x}\to\boldsymbol{\xi}}$ is defined at least on the interval $[0,1]$. This allows us to uniquely define the exponential map $\exp_\mathbf{x} : \mathcal{D}_\mathbf{x} \to \mathcal{M}$ such that $\exp_\mathbf{x}(\boldsymbol{\xi}) = \gamma_{\mathbf{x}\to\boldsymbol{\xi}}(1)$.

The manifold $\mathcal{Q}_\beta^{p,q}$ is geodesically complete, the domain of its exponential map is then $\mathcal{D}_\mathbf{x} = T_\mathbf{x}\mathcal{Q}_\beta^{p,q}$. Using Eq. (5) with $t = 1$, we obtain an exponential map of the entire tangent space to the manifold:

$$\forall\boldsymbol{\xi}\in T_\mathbf{x}\mathcal{Q}_\beta^{p,q}, \ \exp_\mathbf{x}(\boldsymbol{\xi}) = \gamma_{\mathbf{x}\to\boldsymbol{\xi}}(1).$$ (6)

We make the important observation that the image of the exponential map does not necessarily cover the entire manifold: not all points on a manifold are connected by a geodesic. This is the case for our pseudo-hyperboloids. Namely, for a given point $\mathbf{x}\in\mathcal{Q}_\beta^{p,q}$ there exist points $\mathbf{y}$ that are not in the image of the exponential map (*i.e.* there does not exist a tangent vector $\boldsymbol{\xi}$ such that $\mathbf{y} = \exp_\mathbf{x}(\boldsymbol{\xi})$).

**Logarithm map:** We provide a closed-form expression of the logarithm map for pseudo-hyperboloids. Let $\mathcal{U}_\mathbf{x} \subseteq \mathcal{Q}_\beta^{p,q}$ be some neighborhood of $\mathbf{x}$. The logarithm map $\log_\mathbf{x} : \mathcal{U}_\mathbf{x} \to T_\mathbf{x}\mathcal{Q}_\beta^{p,q}$ is defined as the inverse of the exponential map on $\mathcal{U}_\mathbf{x}$ (*i.e.* $\log_\mathbf{x} = \exp_\mathbf{x}^{-1}$). We propose:

$$\forall\mathbf{y}\in\mathcal{U}_\mathbf{x}, \ \log_\mathbf{x}(\mathbf{y}) = \begin{cases} \frac{\cosh^{-1}(\frac{\langle\mathbf{x},\mathbf{y}\rangle_q}{\beta})}{\sqrt{(\frac{\langle\mathbf{x},\mathbf{y}\rangle_q}{\beta})^2-1}}\left(\mathbf{y} - \frac{\langle\mathbf{x},\mathbf{y}\rangle_q}{\beta}\mathbf{x}\right) & \text{if } \frac{\langle\mathbf{x},\mathbf{y}\rangle_q}{|\beta|} < -1 \\ \mathbf{y} - \mathbf{x} & \text{if } \frac{\langle\mathbf{x},\mathbf{y}\rangle_q}{|\beta|} = -1 \\ \frac{\cos^{-1}(\frac{\langle\mathbf{x},\mathbf{y}\rangle_q}{\beta})}{\sqrt{1-(\frac{\langle\mathbf{x},\mathbf{y}\rangle_q}{\beta})^2}}\left(\mathbf{y} - \frac{\langle\mathbf{x},\mathbf{y}\rangle_q}{\beta}\mathbf{x}\right) & \text{if } \frac{\langle\mathbf{x},\mathbf{y}\rangle_q}{|\beta|} \in (-1,1) \end{cases}$$ (7)

By substituting $\boldsymbol{\xi} = \log_\mathbf{x}(\mathbf{y})$ into Eq. (6), one can verify that our formulas are the inverse of the exponential map. The set $\mathcal{U}_\mathbf{x} = \left\{\mathbf{y}\in\mathcal{Q}_\beta^{p,q} : \langle\mathbf{x},\mathbf{y}\rangle_q < |\beta|\right\}$ is called a normal neighborhood of $\mathbf{x}\in\mathcal{Q}_\beta^{p,q}$ since for all $\mathbf{y}\in\mathcal{U}_\mathbf{x}$, there exists a geodesic from $\mathbf{x}$ to $\mathbf{y}$ such that $\log_\mathbf{x}(\mathbf{y}) = \gamma'_{\mathbf{x}\to\log_\mathbf{x}(\mathbf{y})}(0)$. We show in the supp. material that the logarithm map is not defined if $\langle\mathbf{x},\mathbf{y}\rangle_q \geq |\beta|$.

**Proposed dissmilarity:** We define our dissimilarity function based on the general notion of arc length and radius function on pseudo-Riemannian manifolds that we recall in the next paragraph (see details in Chapter 5 of [21]). This corresponds to the geodesic distance in the Riemannian case.

Let $\mathcal{U}_{\mathbf{x}}$ be a normal neighborhood of $\mathbf{x} \in \mathcal{M}$ with $\mathcal{M}$ pseudo-Riemannian. The *radius function* $r_{\mathbf{x}} : \mathcal{U}_{\mathbf{x}} \to \mathbb{R}$ is defined as $r_{\mathbf{x}}(\mathbf{y}) = \sqrt{|g_{\mathbf{x}}\left(\log_{\mathbf{x}}(\mathbf{y}), \log_{\mathbf{x}}(\mathbf{y})\right)|}$ where $g_{\mathbf{x}}$ is the metric at $\mathbf{x}$. If $\sigma_{\mathbf{x} \to \boldsymbol{\xi}}$ is the radial geodesic from $\mathbf{x}$ to $\mathbf{y} \in \mathcal{U}_{\mathbf{x}}$ (*i.e.* $\boldsymbol{\xi} = \log_{\mathbf{x}}(\mathbf{y})$), then the arc length of $\sigma_{\mathbf{x} \to \boldsymbol{\xi}}$ equals $r_{\mathbf{x}}(\mathbf{y})$.

We then define the geodesic "distance" between $\mathbf{x} \in \mathcal{Q}_{\beta}^{p,q}$ and $\mathbf{y} \in \mathcal{U}_{\mathbf{x}}$ as the arc length of $\sigma_{\mathbf{x} \to \log_{\mathbf{x}}(\mathbf{y})}$:

$$\mathsf{d}_{\gamma}(\mathbf{x}, \mathbf{y}) = \sqrt{|\|\log_{\mathbf{x}}(\mathbf{y})\|_q^2|} = \begin{cases} \sqrt{|\beta|} \cosh^{-1}\left(\frac{\langle \mathbf{x}, \mathbf{y} \rangle_q}{\beta}\right) & \text{if } \frac{\langle \mathbf{x}, \mathbf{y} \rangle_q}{|\beta|} < -1 \\ 0 & \text{if } \frac{\langle \mathbf{x}, \mathbf{y} \rangle_q}{|\beta|} = -1 \\ \sqrt{|\beta|} \cos^{-1}\left(\frac{\langle \mathbf{x}, \mathbf{y} \rangle_q}{\beta}\right) & \text{if } \frac{\langle \mathbf{x}, \mathbf{y} \rangle_q}{|\beta|} \in (-1, 1) \end{cases} \tag{8}$$

It is important to note that our "distance" is **not** a distance metric. However, it satisfies the axioms of a *symmetric premetric*: (i) $\mathsf{d}_{\gamma}(\mathbf{x}, \mathbf{y}) = \mathsf{d}_{\gamma}(\mathbf{y}, \mathbf{x}) \geq 0$ and (ii) $\mathsf{d}_{\gamma}(\mathbf{x}, \mathbf{x}) = 0$. These conditions are sufficient to quantify the notion of nearness via a $\rho$-ball centered at $\mathbf{x}$: $B_{\mathbf{x}}^{\rho} = \{\mathbf{y} : \mathsf{d}_{\gamma}(\mathbf{x}, \mathbf{y}) < \rho\}$.

In general, topological spaces provide a qualitative (not necessarily quantitative) way to detect "nearness" through the concept of a neighborhood at a point [15]. Something is true "near $\mathbf{x}$" if it is true in the neighborhood of $\mathbf{x}$ (*e.g.* in $B_{\mathbf{x}}^{\rho}$). Our premetric is similar to metric learning methods [13, 14, 29] that learn a Mahalanobis-like distance pseudo-metric parameterized by a positive semi-definite matrix. Pairs of distinct points can have zero "distance" if the matrix is not positive definite. However, unlike classic metric learning, we can have triplets $(\mathbf{x}, \mathbf{y}, \mathbf{z})$ that satisfy $\mathsf{d}_{\gamma}(\mathbf{x}, \mathbf{y}) = \mathsf{d}_{\gamma}(\mathbf{x}, \mathbf{z}) = 0$ but $\mathsf{d}_{\gamma}(\mathbf{y}, \mathbf{z}) > 0$ (*e.g.* $\mathbf{x} = (1,0,0,0)^{\top}, \mathbf{y} = (1,1,1,0)^{\top}, \mathbf{z} = (1,1,0,1)^{\top}$ in $\mathcal{Q}_{-1}^{2,1}$).

Since the logarithm map is not defined if $\langle \mathbf{x}, \mathbf{y} \rangle_q \geq |\beta|$, we propose to use the following continuous approximation defined on the whole manifold instead:

$$\forall \mathbf{x} \in \mathcal{Q}_{\beta}^{p,q}, \mathbf{y} \in \mathcal{Q}_{\beta}^{p,q}, \ \mathsf{D}_{\gamma}(\mathbf{x}, \mathbf{y}) = \begin{cases} \mathsf{d}_{\gamma}(\mathbf{x}, \mathbf{y}) & \text{if } \langle \mathbf{x}, \mathbf{y} \rangle_q \leq 0 \\ \sqrt{|\beta|}\left(\frac{\pi}{2} + \frac{\langle \mathbf{x}, \mathbf{y} \rangle_q}{|\beta|}\right) & \text{otherwise} \end{cases} \tag{9}$$

To the best of our knowledge, the explicit formulation of the logarithm map for $\mathcal{Q}_{\beta}^{p,q}$ in Eq. (7) and its corresponding radius function in Eq. (8) to define a dissimilarity function are novel. We have also proposed some linear approximation to evaluate dissimilarity when the logarithm map is not defined but other choices are possible. For instance, when a geodesic does not exist, a standard way in differential geometry to calculate curves is to consider broken geodesics. One might consider instead the dissimilarity $\mathsf{d}_{\gamma}(\mathbf{x}, -\mathbf{x}) + \mathsf{d}_{\gamma}(-\mathbf{x}, \mathbf{y}) = \pi\sqrt{|\beta|} + \mathsf{d}_{\gamma}(-\mathbf{x}, \mathbf{y})$ if $\log_{\mathbf{x}}(\mathbf{y})$ is not defined since $-\mathbf{x} \in \mathcal{Q}_{\beta}^{p,q}$ and $\log_{-\mathbf{x}}(\mathbf{y})$ is defined. This interesting problem is left for future research.

## 4 Ultrahyperbolic Optimization

In this section we present optimization frameworks to optimize any differentiable function defined on $\mathcal{Q}_{\beta}^{p,q}$. Our goal is to compute descent directions on the ultrahyperbolic manifold. We consider two approaches. In the first approach, we map our representation from Euclidean space to ultrahyperbolic space. This is similar to the approach taken by [11] in hyperbolic space. In the second approach, we optimize using gradients defined directly in pseudo-Riemannian tangent space. We propose a novel descent direction which guarantees the minimization of some cost function.

### 4.1 Euclidean optimization via a differentiable mapping onto $\mathcal{Q}_{\beta}^{p,q}$

Our first method maps Euclidean representations that lie in $\mathbb{R}^d$ to the pseudo-hyperboloid $\mathcal{Q}_{\beta}^{p,q}$, and the chain rule is exploited to perform standard gradient descent. To this end, we construct a differentiable mapping $\varphi : \mathbb{R}_*^{q+1} \times \mathbb{R}^p \to \mathcal{Q}_{\beta}^{p,q}$. The image of a point already on $\mathcal{Q}_{\beta}^{p,q}$ under the mapping $\varphi$ is itself: $\forall \mathbf{x} \in \mathcal{Q}_{\beta}^{p,q}, \varphi(\mathbf{x}) = \mathbf{x}$. Let $\mathcal{S}^q = \left\{\mathbf{x} \in \mathbb{R}^{q+1} : \|\mathbf{x}\| = 1\right\}$ denote the unit $q$-sphere. We first introduce the following diffeomorphisms:

**Theorem 4.1** (Diffeomorphisms). *For any $\beta < 0$, there is a diffeomorphism $\psi : \mathcal{Q}_\beta^{p,q} \to \mathcal{S}^q \times \mathbb{R}^p$.*

*Let us note $\mathbf{x} = \begin{pmatrix} \mathbf{t} \\ \mathbf{s} \end{pmatrix} \in \mathcal{Q}_\beta^{p,q}$ with $\mathbf{t} \in \mathbb{R}_*^{q+1}$ and $\mathbf{s} \in \mathbb{R}^p$, let us note $\mathbf{z} = \begin{pmatrix} \mathbf{u} \\ \mathbf{v} \end{pmatrix} \in \mathcal{S}^q \times \mathbb{R}^p$ where $\mathbf{u} \in \mathcal{S}^q$ and $\mathbf{v} \in \mathbb{R}^p$. The mapping $\psi$ and its inverse $\psi^{-1}$ are formulated (see proofs in supp. material):*

$$\psi(\mathbf{x}) = \begin{pmatrix} \frac{1}{\|\mathbf{t}\|}\mathbf{t} \\ \frac{1}{\sqrt{|\beta|}}\mathbf{s} \end{pmatrix} \quad and \quad \psi^{-1}(\mathbf{z}) = \sqrt{|\beta|}\begin{pmatrix} \sqrt{1 + \|\mathbf{v}\|^2}\mathbf{u} \\ \mathbf{v} \end{pmatrix}. \quad (10)$$

With these mappings, any vector $\mathbf{x} \in \mathbb{R}_*^{q+1} \times \mathbb{R}^p$ can be mapped to $\mathcal{Q}_\beta^{p,q}$ via $\varphi = \psi^{-1} \circ \psi$. $\varphi$ is differentiable everywhere except when $x_0 = \cdots = x_q = 0$, which should never occur in practice. It can therefore be optimized using standard gradient methods.

## 4.2 Pseudo-Riemannian optimization

We now introduce a novel method to optimize any differentiable function $f : \mathcal{Q}_\beta^{p,q} \to \mathbb{R}$ defined on the pseudo-hyperboloid. As we show below, the (negative of the) pseudo-Riemannian gradient is not a descent direction. We propose a simple and efficient way to calculate a descent direction.

**Pseudo-Riemannian gradient:** Since $\mathbf{x} \in \mathcal{Q}_\beta^{p,q}$ also lies in the Euclidean ambient space $\mathbb{R}^d$, the function $f$ has a well defined Euclidean gradient $\nabla f(\mathbf{x}) = (\partial f(\mathbf{x})/\partial x_0, \cdots, \partial f(\mathbf{x})/\partial x_{p+q})^\top \in \mathbb{R}^d$. The gradient of $f$ in the pseudo-Euclidean ambient space $\mathbb{R}^{p,q+1}$ is $(\mathbf{G}^{-1}\nabla f(\mathbf{x})) = (\mathbf{G}\nabla f(\mathbf{x})) \in \mathbb{R}^{p,q+1}$. Since $\mathcal{Q}_\beta^{p,q}$ is a submanifold of $\mathbb{R}^{p,q+1}$, the pseudo-Riemannian gradient $Df(\mathbf{x}) \in T_\mathbf{x}\mathcal{Q}_\beta^{p,q}$ of $f$ on $\mathcal{Q}_\beta^{p,q}$ is the orthogonal projection of $(\mathbf{G}\nabla f(\mathbf{x}))$ onto $T_\mathbf{x}\mathcal{Q}_\beta^{p,q}$ (see Chapter 4 of [21]):

$$Df(\mathbf{x}) = \Pi_\mathbf{x}(\mathbf{G}\nabla f(\mathbf{x})) = \mathbf{G}\nabla f(\mathbf{x}) - \frac{\langle \mathbf{G}\nabla f(\mathbf{x}), \mathbf{x}\rangle_q}{\langle \mathbf{x}, \mathbf{x}\rangle_q}\mathbf{x} = \mathbf{G}\nabla f(\mathbf{x}) - \frac{\langle \nabla f(\mathbf{x}), \mathbf{x}\rangle}{\langle \mathbf{x}, \mathbf{x}\rangle_q}\mathbf{x}. \quad (11)$$

This gradient forms the foundation of our descent method optimizer as will be shown in Eq. (13).

**Iterative optimization:** Our goal is to iteratively decrease the value of the function $f$ by following some descent direction. Since $\mathcal{Q}_\beta^{p,q}$ is not a vector space, we do not "follow the descent direction" by adding the descent direction multiplied by a step size as this would result in a new point that does not necessarily lie on $\mathcal{Q}_\beta^{p,q}$. Instead, to remain on the manifold, we use our exponential map defined in Eq. (6). This is a standard way to optimize on Riemannian manifolds [1]. Given a step size $t > 0$, one step of descent along a tangent vector $\boldsymbol{\zeta} \in T_\mathbf{x}\mathcal{Q}_\beta^{p,q}$ is given by:

$$\mathbf{y} = \exp_\mathbf{x}(t\boldsymbol{\zeta}) \in \mathcal{Q}_\beta^{p,q}. \quad (12)$$

**Descent direction:** We now explain why the negative of the pseudo-Riemannian gradient is not a descent direction. Our explanation extends Chapter 3 of [20] that gives the criteria for a tangent vector $\boldsymbol{\zeta}$ to be a descent direction when the domain of the optimized function is a Euclidean space. By using the properties described in Section 3, we know that for all $t \in \mathbb{R}$ and all $\boldsymbol{\xi} \in T_\mathbf{x}\mathcal{Q}_\beta^{p,q}$, we have the equalities: $\exp_\mathbf{x}(t\boldsymbol{\xi}) = \gamma_{\mathbf{x}\to t\boldsymbol{\xi}}(1) = \gamma_{\mathbf{x}\to\boldsymbol{\xi}}(t)$ so we can equivalently fix $t$ to 1 and choose the scale of $\boldsymbol{\xi}$ appropriately. By exploiting Taylor's first-order approximation, there exists some small enough tangent vector $\boldsymbol{\zeta} \neq \mathbf{0}$ (*i.e.* with $\exp_\mathbf{x}(\boldsymbol{\zeta})$ belonging to a convex neighborhood of $\mathbf{x}$ [4, 8]) that satisfies the following conditions: $\gamma_{\mathbf{x}\to\boldsymbol{\zeta}}(0) = \mathbf{x} \in \mathcal{Q}_\beta^{p,q}$, $\gamma'_{\mathbf{x}\to\boldsymbol{\zeta}}(0) = \boldsymbol{\zeta} \in T_\mathbf{x}\mathcal{Q}_\beta^{p,q}$, $\gamma_{\mathbf{x}\to\boldsymbol{\zeta}}(1) = \mathbf{y} \in \mathcal{Q}_\beta^{p,q}$, and the function $f \circ \gamma_{\mathbf{x}\to\boldsymbol{\zeta}} : \mathbb{R} \to \mathbb{R}$ can be approximated at $t = 1$ by:

$$f(\mathbf{y}) = f \circ \gamma_{\mathbf{x}\to\boldsymbol{\zeta}}(1) \simeq f \circ \gamma_{\mathbf{x}\to\boldsymbol{\zeta}}(0) + (f \circ \gamma_{\mathbf{x}\to\boldsymbol{\zeta}})'(0) = f(\mathbf{x}) + \langle Df(\mathbf{x}), \boldsymbol{\zeta}\rangle_q. \quad (13)$$

where we use the following properties: $\forall t, (f \circ \gamma)'(t) = \mathrm{d}f(\gamma'(t)) = g_{\gamma(t)}(Df(\gamma(t)), \gamma'(t))$ (see details in pages 11, 15 and 85 of [21]), $\mathrm{d}f$ is the differential of $f$ and $\gamma$ is a geodesic.

To be a descent direction at $\mathbf{x}$ (*i.e.* so that $f(\mathbf{y}) < f(\mathbf{x})$), the search direction $\boldsymbol{\zeta}$ has to satisfy $\langle Df(\mathbf{x}), \boldsymbol{\zeta}\rangle_q < 0$. However, choosing $\boldsymbol{\zeta} = -\eta Df(\mathbf{x})$, where $\eta > 0$ is a step size, might increase the function value if the scalar product $\langle \cdot, \cdot \rangle_q$ is not positive definite. If $p + q \geq 1$, then $\langle \cdot, \cdot \rangle_q$ is positive definite only if $q = 0$ (see details in supp. material), and it is negative definite iff $p = 0$ since $\langle \cdot, \cdot \rangle_q = -\langle \cdot, \cdot \rangle$ in this case. A simple solution would be to choose $\boldsymbol{\zeta} = \pm\eta Df(\mathbf{x})$ depending on the sign of $\langle Df(\mathbf{x}), \boldsymbol{\zeta}\rangle_q$, but $\langle Df(\mathbf{x}), \boldsymbol{\zeta}\rangle_q$ might be equal to 0 even if $Df(\mathbf{x}) \neq \mathbf{0}$ if $\langle \cdot, \cdot \rangle_q$ is indefinite. The optimization algorithm might then be stuck to a level set of $f$, which is problematic.

---

**Algorithm 1** Pseudo-Riemannian optimization on $\mathcal{Q}_\beta^{p,q}$

---

**input:** differentiable function $f : \mathcal{Q}_\beta^{p,q} \to \mathbb{R}$ to be minimized, some initial value of $\mathbf{x} \in \mathcal{Q}_\beta^{p,q}$

1: **while** not converge **do**
2:      Calculate $\nabla f(\mathbf{x})$          $\triangleright$ *i.e.* the Euclidean gradient of $f$ at $\mathbf{x}$ in the Euclidean ambient space
3:      $\chi \leftarrow \Pi_\mathbf{x}(\mathbf{G}\Pi_\mathbf{x}(\mathbf{G}\nabla f(\mathbf{x})))$          $\triangleright$ see Eq. (14)
4:      $\mathbf{x} \leftarrow \exp_\mathbf{x}(-\eta\chi)$          $\triangleright$ where $\eta > 0$ is a step size (*e.g.* determined with line search)
5: **end while**

---

**Proposed solution:** To ensure that $\zeta \in T_\mathbf{x}\mathcal{Q}_\beta^{p,q}$ is a descent direction, we propose a simple expression that satisfies $\langle Df(\mathbf{x}), \zeta\rangle_q < 0$ if $Df(\mathbf{x}) \neq \mathbf{0}$ and $\langle Df(\mathbf{x}), \zeta\rangle_q = 0$ otherwise. We propose to formulate $\zeta = -\eta\Pi_\mathbf{x}(\mathbf{G}Df(\mathbf{x})) \in T_\mathbf{x}\mathcal{Q}_\beta^{p,q}$, and we define the following tangent vector $\chi = -\frac{1}{\eta}\zeta$:

$$\chi = \Pi_\mathbf{x}(\mathbf{G}Df(\mathbf{x})) = \nabla f(\mathbf{x}) - \frac{\langle\nabla f(\mathbf{x}), \mathbf{x}\rangle}{\langle\mathbf{x}, \mathbf{x}\rangle_q}\mathbf{G}\mathbf{x} - \frac{\langle\nabla f(\mathbf{x}), \mathbf{x}\rangle_q}{\langle\mathbf{x}, \mathbf{x}\rangle_q}\mathbf{x} + \frac{\|\mathbf{x}\|^2\langle\nabla f(\mathbf{x}), \mathbf{x}\rangle}{\langle\mathbf{x}, \mathbf{x}\rangle_q^2}\mathbf{x}. \quad (14)$$

The tangent vector $\zeta$ is a descent direction because $\langle Df(\mathbf{x}), \zeta\rangle_q = -\eta\langle Df(\mathbf{x}), \chi\rangle_q$ is nonpositive:

$$\langle Df(\mathbf{x}), \chi\rangle_q = \|\nabla f(\mathbf{x})\|^2 - 2\frac{\langle\nabla f(\mathbf{x}), \mathbf{x}\rangle\langle\nabla f(\mathbf{x}), \mathbf{x}\rangle_q}{\langle\mathbf{x}, \mathbf{x}\rangle_q} + \frac{\langle\nabla f(\mathbf{x}), \mathbf{x}\rangle^2\|\mathbf{x}\|^2}{\langle\mathbf{x}, \mathbf{x}\rangle_q^2} \quad (15)$$

$$= \|\mathbf{G}\nabla f(\mathbf{x}) - \frac{\langle\nabla f(\mathbf{x}), \mathbf{x}\rangle}{\langle\mathbf{x}, \mathbf{x}\rangle_q}\mathbf{x}\|^2 = \|Df(\mathbf{x})\|^2 \geq 0. \quad (16)$$

We also have $\langle Df(\mathbf{x}), \chi\rangle_q = \|Df(\mathbf{x})\|^2 = 0$ iff $Df(\mathbf{x}) = \mathbf{0}$ (*i.e.* $\mathbf{x}$ is a stationary point). It is worth noting that $Df(\mathbf{x}) = \mathbf{0}$ implies $\chi = \Pi_\mathbf{x}(\mathbf{G}\mathbf{0}) = \mathbf{0}$. Moreover, $\chi = \mathbf{0}$ implies that $\|Df(\mathbf{x})\|^2 = \langle Df(\mathbf{x}), \mathbf{0}\rangle_q = 0$. We then have $\chi = \mathbf{0}$ iff $Df(\mathbf{x}) = \mathbf{0}$.

Our proposed algorithm to the minimization problem $\min_{\mathbf{x}\in\mathcal{Q}_\beta^{p,q}} f(\mathbf{x})$ is illustrated in Algorithm 1. Following generic Riemannian optimization algorithms [1], at each iteration, it first computes the descent direction $-\chi \in T_\mathbf{x}\mathcal{Q}_\beta^{p,q}$, then decreases the function by applying the exponential map defined in Eq. (6). It is worth noting that our proposed descent method can be applied to any differentiable function $f : \mathcal{Q}_\beta^{p,q} \to \mathbb{R}$, not only to those that exploit the distance introduced in Section 3.

Interestingly, our method can also be seen as a preconditioning technique [20] where the descent direction is obtained by preconditioning the pseudo-Riemannian gradient $Df(\mathbf{x})$ with the matrix $\mathbf{P}_\mathbf{x} = \left[\mathbf{G} - \frac{1}{\langle\mathbf{x},\mathbf{x}\rangle_q}\mathbf{x}\mathbf{x}^\top\right] \in \mathbb{R}^{d\times d}$. In other words, we have $\chi = \mathbf{P}_\mathbf{x}Df(\mathbf{x}) = \Pi_\mathbf{x}(\mathbf{G}Df(\mathbf{x}))$.

In the more general setting of pseudo-Riemannian manifolds, another preconditioning technique was proposed in [8]. The method in [8] requires performing a Gram-Schmidt process at each iteration to obtain an (*ordered* [28]) orthonormal basis of the tangent space at $\mathbf{x}$ *w.r.t.* the induced quadratic form of the manifold. However, the Gram-Schmidt process is unstable and has algorithmic complexity that is cubic in the dimensionality of the tangent space. On the other hand, our method is more stable and its algorithmic complexity is linear in the dimensionality of the tangent space.

## 5 Experiments

We now experimentally validate our proposed optimization methods and the effectiveness of our dissimilarity function. Our main experimental results can be summarized as follows:

• Both optimizers introduced in Section 4 decrease some objective function $f : \mathcal{Q}_\beta^{p,q} \to \mathbb{R}$. While both optimizers manage to learn high-dimensional representations that satisfy the problem-dependent training constraints, only the pseudo-Riemannian optimizer satisfies all the constraints in lower-dimensional spaces. This is because it exploits the underlying metric of the manifold.

• Hyperbolic representations are popular in machine learning as they are well suited to represent hierarchical trees [10, 18, 19]. On the other hand, hierarchical datasets whose graph contains cycles cannot be represented using trees. Therefore, we propose to represent such graphs using our ultrahyperbolic representations. An important example are community graphs such as Zachary's karate club [30] that contain leaders. Because our ultrahyperbolic representations are more flexible

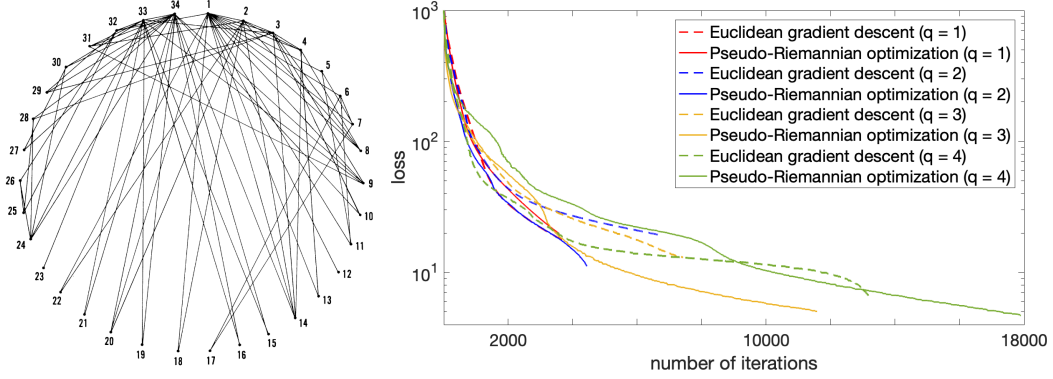

Figure 3: (left) graphic representation of Zachary's karate club (figure courtesy of [30]). (right) Loss values of Eq. (17) for different numbers of time dimensions and optimizers.

than hyperbolic representations, we believe that our representations are better suited for these non tree-like hierarchical structures.

**Graph:** Our ultrahyperbolic representations describe graph-structured datasets. Each dataset is an undirected weighted graph $G = (V, E)$ which has node-set $V = \{v_i\}_{i=1}^n$ and edge-set $E = \{e_k\}_{k=1}^m$. Each edge $e_k$ is weighted by an arbitrary capacity $c_k \in \mathbb{R}_+$ that models the strength of the relationship between nodes. The higher the capacity $c_k$, the stronger the relationship between the nodes connected by $e_k$.

**Learned representations:** Our problem formulation is inspired by hyperbolic representation learning approaches [18, 19] where the nodes of a tree (*i.e.* graph without cycles) are represented in hyperbolic space. The hierarchical structure of the tree is then reflected by the order of distances between its nodes. More precisely, a node representation is learned so that each node is closer to its descendants and ancestors in the tree (*w.r.t.* the hyperbolic distance) than to any other node. For example, in a hierarchy of words, ancestors and descendants are hypernyms and hyponyms, respectively.

Our goal is to learn a set of $n$ points $\mathbf{x}_1, \cdots, \mathbf{x}_n \in \mathcal{Q}_\beta^{p,q}$ (embeddings) from a given graph $G$. The presence of cycles in the graph makes it difficult to determine ancestors and descendants. For this reason, we introduce for each pair of nodes $(v_i, v_j) = e_k \in E$, the set of "*weaker*" pairs that have lower capacity: $\mathcal{W}(e_k) = \{e_l : c_k > c_l\} \cup \{(v_a, v_b) : (v_a, v_b) \notin E\}$. Our goal is to learn representations such that pairs $(v_i, v_j)$ with higher capacity have their representations $(\mathbf{x}_i, \mathbf{x}_j)$ closer to each other than weaker pairs. Following [18], we formulate our problem as:

$$\min_{\mathbf{x}_1, \cdots, \mathbf{x}_n \in \mathcal{Q}_\beta^{p,q}} \sum_{(v_i, v_j) = e_k \in E} - \log \frac{\exp\left(-\mathsf{d}(\mathbf{x}_i, \mathbf{x}_j)/\tau\right)}{\sum_{(v_a, v_b) \in \mathcal{W}(e_k) \cup \{e_k\}} \exp\left(-\mathsf{d}(\mathbf{x}_a, \mathbf{x}_b)/\tau\right)} \tag{17}$$

where d is the chosen dissimilarity function (*e.g.* $\mathsf{D}_\gamma(\cdot, \cdot)$ defined in Eq. (9)) and $\tau > 0$ is a fixed temperature parameter. The formulation of Eq. (17) is classic in the metric learning literature [3, 12, 27] and corresponds to optimizing some order on the learned distances via a softmax function.

**Implementation details:** We coded our approach in PyTorch [22] that automatically calculates the Euclidean gradient $\nabla f(\mathbf{x}_i)$. Initially, a random set of vectors $\{\mathbf{z}_i\}_{i=1}^n$ is generated close to the positive pole $(\sqrt{|\beta|}, 0, \cdots, 0) \in \mathcal{Q}_\beta^{p,q}$ with every coordinate perturbed uniformly with a random value in the interval $[-\varepsilon, \varepsilon]$ where $\varepsilon > 0$ is chosen small enough so that $\|\mathbf{z}_i\|_q^2 < 0$. We set $\beta = -1$, $\varepsilon = 0.1$ and $\tau = 10^{-2}$. Initial embeddings are generated as follows: $\forall i, \mathbf{x}_i = \sqrt{|\beta|} \frac{\mathbf{z}_i}{\sqrt{|\|\mathbf{z}_i\|_q^2|}} \in \mathcal{Q}_\beta^{p,q}$.

**Zachary's karate club dataset** [30] is a social network graph of a karate club comprised of $n = 34$ nodes, each representing a member of the karate club. The club was split due to a conflict between instructor "Mr. Hi" (node $v_1$) and administrator "John A" (node $v_n$). The remaining members now have to decide whether to join the new club created by $v_1$ or not. In [30], Zachary defines a matrix of relative strengths of the friendships in the karate club called $\mathbf{C} \in \{0, 1, \cdots, 7\}^{n \times n}$ and that depends on various criteria. We note that the matrix is not symmetric and has 7 different pairs $(v_i, v_j)$ for which $\mathbf{C}_{ij} \neq \mathbf{C}_{ji}$. Since our dissimilarity function is symmetric, we consider the symmetric matrix

Table 1: Evaluation scores for the different learned representations (mean $\pm$ standard deviation)

| Evaluation metric | $\mathbb{R}^4$ (flat) | $\mathcal{Q}_{-1}^{4,0}$ (hyperbolic) | $\mathcal{Q}_{-1}^{3,1}$ (ours) | $\mathcal{Q}_{-1}^{2,2}$ (ours) | $\mathcal{Q}_{-1}^{1,3}$ (ours) | $\mathcal{Q}_{-1}^{0,4}$ (spherical) |
|---|---|---|---|---|---|---|
| Rank of the first leader | $5.4 \pm 1.1$ | $2.8 \pm 0.4$ | $\mathbf{1.2 \pm 0.4}$ | $\mathbf{1.2 \pm 0.4}$ | $1.8 \pm 0.8$ | $2.0 \pm 0.7$ |
| Rank of the second leader | $6.6 \pm 0.9$ | $4.2 \pm 0.7$ | $\mathbf{2.4 \pm 0.9}$ | $2.6 \pm 0.5$ | $4.0 \pm 1.2$ | $4.0 \pm 1.4$ |
| top 5 Spearman's $\rho$ | $-0.44 \pm 0.19$ | $0.20 \pm 0.48$ | $\mathbf{0.76 \pm 0.21}$ | $0.66 \pm 0.30$ | $0.36 \pm 0.40$ | $0.18 \pm 0.37$ |
| top 10 Spearman's $\rho$ | $0.00 \pm 0.14$ | $0.38 \pm 0.06$ | $0.74 \pm 0.11$ | $\mathbf{0.79 \pm 0.12}$ | $0.71 \pm 0.08$ | $0.55 \pm 0.20$ |

$\mathbf{S} = \mathbf{C} + \mathbf{C}^{\top}$ instead. The value of $\mathbf{S}_{ij}$ is the capacity/weight assigned to the edge joining $v_i$ and $v_j$, and there is no edge between $v_i$ and $v_j$ if $\mathbf{S}_{ij} = 0$. Fig. 3 (left) illustrates the 34 nodes of the dataset, an edge joining the nodes $v_i$ and $v_j$ is drawn iff $\mathbf{S}_{ij} \neq 0$. The level of a node in the hierarchy corresponds approximately to its height in the figure.

**Optimizers:** We validate that our optimizers introduced in Section 4 decrease the cost function. First, we consider the simple unweighted case where every edge weight is 1. For each edge $e_k \in E$, $\mathcal{W}(e_k)$ is then the set of pairs of nodes that are not connected. In other words, Eq. (17) learns node representations that have the property that every connected pair of nodes has smaller distance than non-connected pairs. We use this condition as a stopping criterion of our algorithm.

Fig. 3 (right) illustrates the loss values of Eq. (17) as a function of the number of iterations with the Euclidean gradient descent (Section 4.1) and our pseudo-Riemannian optimizer (introduced in Section 4.2). In each test, we vary the number of time dimensions $q + 1$ while the ambient space is of fixed dimensionality $d = p + q + 1 = 10$. We omit the case $q = 0$ since it corresponds to the (hyperbolic) Riemannian case already considered in [11, 19]. Both optimizers decrease the function and manage to satisfy all the expected distance relations. We note that when we use $-Df(\mathbf{x})$ instead of $-\chi$ as a search direction, the algorithm does not converge. Moreover, our pseudo-Riemannian optimizer manages to learn representations that satisfy all the constraints for low-dimensional manifolds such as $\mathcal{Q}_{-1}^{4,1}$ and $\mathcal{Q}_{-1}^{4,2}$, while the optimizer introduced in Section 4.1 does not. Consequently, we only use the pseudo-Riemannian optimizer in the following results.

**Hierarchy extraction:** To quantitatively evaluate our approach, we apply it to the problem of predicting the high-level nodes in the hierarchy from the weighted matrix $\mathbf{S}$ given as supervision. We consider the challenging low-dimensional setting where all the learned representations lie on a 4-dimensional manifold (*i.e.* $p + q + 1 = 5$). Hyperbolic distances are known to grow exponentially as we get further from the origin. Therefore, the sum of distances $\delta_i = \sum_{j=1}^{n} \mathsf{d}(\mathbf{x}_i, \mathbf{x}_j)$ of a node $v_i$ with all other nodes is a good indication of importance. Intuitively, high-level nodes will be closer to most nodes than low-level nodes. We then sort the scores $\delta_1, \cdots, \delta_n$ in ascending order and report the ranks of the two leaders $v_1$ or $v_n$ (in no particular order) in the first two rows of Table 1 averaged over 5 different initializations/runs. Leaders tend to have a smaller $\delta_i$ score with ultrahyperbolic distances than with Euclidean, hyperbolic or spherical distances.

Instead of using $\delta_i$ for hyperbolic representations, the importance of a node $v_i$ can be evaluated by using the Euclidean norm of its embedding $\mathbf{x}_i$ as proxy [11, 18, 19]. This is because high-level nodes of a tree in hyperbolic space are usually closer to the origin than low-level nodes. Not surprisingly, this proxy leads to worse performance ($8.6 \pm 2.3$ and $18.6 \pm 4.9$) as the relationships are not that of a tree. Since hierarchy levels are hard to compare for low-level nodes, we select the 10 (or 5) most influential members based on the score $s_i = \sum_{j=1}^{n} \mathbf{S}_{ij}$. The corresponding nodes are 34, 1, 33, 3, 2, 32, 24, 4, 9, 14 (in that order). Spearman's rank correlation coefficient [24] between the selected scores $s_i$ and corresponding $\delta_i$ is reported in Table 1 and shows the relevance of our representations.

Due to lack of space, we also report in the supp. material similar experiments on a larger hierarchical dataset [9] that describes co-authorship from papers published at NIPS from 1988 to 2003.

## 6 Conclusion

We have introduced ultrahyperbolic representations. Our representations lie on a pseudo-Riemannian manifold of constant nonzero curvature which generalizes hyperbolic and spherical geometries and includes them as submanifolds. Any relationship described in those geometries can then be described with our representations that are more flexible. We have introduced new optimization tools and experimentally shown that our representations can extract hierarchies in graphs that contain cycles.

## Broader Impact

We introduce a novel way of representing relationships between data points by considering the geometry of non-Riemannian manifolds of constant nonzero curvature. The relationships between data points are described by a dissimilarity function that we introduce and exploits the structure of the manifold. It is more flexible than the distance metric used in hyperbolic and spherical geometries often used in machine learning and computer vision. Nonetheless, since the problems involving our representations are not straightforward to optimize, we propose novel optimization algorithms that can potentially benefit the machine learning, computer vision and natural language processing communities. Indeed, our method is application agnostic and could extend existing frameworks.

Our contribution is mainly theoretical but we have included one practical application. Similarly to hyperbolic representations that are popular for representing tree-like data, we have shown that our representations are well adapted to the more general case of hierarchical graphs with cycles. These graphs appear in many different fields of research such as medicine, molecular biology and the social sciences. For example, an ultrahyperbolic representation of proteins might assist in understanding their complicated folding mechanisms. Moreover, these representations could assist in analyzing features of social media such as discovering new trends and leading "connectors". The impact of community detection for commercial or political advertising is already known in social networking services. We foresee that our method will have many more graph-based practical applications.

We know of very few applications outside of general relativity that use pseudo-Riemannian geometry. We hope that our research will stimulate other applications in machine learning and related fields. Finally, although we have introduced a novel descent direction for our optimization algorithm, future research could study and improve its rate of convergence.

## Acknowledgments and Disclosure of Funding

We thank Jonah Philion, Guojun Zhang and the anonymous reviewers for helpful feedback on early versions of this manuscript.

This article was entirely funded by NVIDIA corporation. Marc Law and Jos Stam completed this working from home during the COVID-19 pandemic.

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
