[Supplementary Material]

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

*: $\mathsf{d}_\gamma(\mathbf{a}, \mathbf{b}) < \mathsf{d}_\gamma(\mathbf{c}, \mathbf{d})$ iff $\mathsf{d}_q(\mathbf{a}, \mathbf{b}) < \mathsf{d}_q(\mathbf{c}, \mathbf{d})$. This relation is satisfied for two special cases of pseudo-hyperboloids for which the geodesic distance is well known:

• **Spherical manifold ($\mathcal{Q}_{\beta}^{0,q}$):** If $p = 0$, the geodesic distance $\mathsf{d}_\gamma(\mathbf{a}, \mathbf{b}) = \sqrt{|\beta|} \cos^{-1}\left(\frac{\langle \mathbf{a}, \mathbf{b} \rangle_q}{\beta}\right)$ is called spherical distance. In practice, the cosine similarity $\frac{\langle \cdot, \cdot \rangle_q}{\beta}$ is often considered instead of $\mathsf{d}_\gamma(\cdot, \cdot)$ since it satisfies $\mathsf{d}_\gamma(\mathbf{a}, \mathbf{b}) < \mathsf{d}_\gamma(\mathbf{c}, \mathbf{d})$ iff $\langle \mathbf{a}, \mathbf{b} \rangle_q < \langle \mathbf{c}, \mathbf{d} \rangle_q$ iff $\mathsf{d}_q(\mathbf{a}, \mathbf{b}) < \mathsf{d}_q(\mathbf{c}, \mathbf{d})$.

• **Hyperbolic manifold (upper sheet of the two-sheet hyperboloid $\mathcal{Q}_{\beta}^{p,0}$):** If $q = 0$, the geodesic distance $\mathsf{d}_\gamma(\mathbf{a}, \mathbf{b}) = \sqrt{|\beta|} \cosh^{-1}\left(\frac{\langle \mathbf{a}, \mathbf{b} \rangle_q}{\beta}\right)$ with $a_0 > 0$ and $b_0 > 0$ is called Poincaré distance [19]. The (extrinsic) Lorentzian distance was shown to be a good proxy in hyperbolic geometry [11].

For the **ultrahyperbolic** case (*i.e.* $q \geq 1$ and $p \geq 2$), the distance relations are not preserved: $\mathsf{d}_\gamma(\mathbf{a}, \mathbf{b}) < \mathsf{d}_\gamma(\mathbf{c}, \mathbf{d}) \not\Longleftrightarrow \mathsf{d}_q(\mathbf{a}, \mathbf{b}) < \mathsf{

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

# A Supplementary material of "Ultrahyperbolic Representation Learning"

The supplementary material is structured as follows:

• In Section B, we provide a short discussion on the choice of geometry to represent graphs.

• In Section C.1, we study the formulation of the geodesics in Eq. (5). It is combination of hyperbolic, spherical and flat space due to its formulation.

• In Section C.2, we explain why $\log_{\mathbf{x}}(\mathbf{y})$ (see Eq. (7)) is not defined if $\langle \mathbf{x}, \mathbf{y} \rangle_q \geq |\beta|$.

• In Section C.3, we explain the anti-isometry between $\mathcal{Q}_\beta^{p,q}$ or $\mathcal{Q}_{-\beta}^{q+1,p-1}$.

• In Section C.4, we study the curvature of $\mathcal{Q}_\beta^{p,q}$.

• In Section C.5, we give the proof of Theorem 4.1.

• In Section C.6, we explain why the upper sheet of the two-sheet hyperboloid $\mathcal{Q}_\beta^{p,0}$ (*i.e.* the case where $q = 0$) is a Riemannian manifold even if its metric matrix $\mathbf{G} = \mathbf{I}_{1,p}$ is not positive definite.

• In Section D, we report additional experiments.

# B    Choice of Geometry

The choice of geometry to represent graphs is still an open problem in general. It depends on the topology of the graph and the kind of relationship between nodes. For instance, hyperbolic geometry was mathematically shown to be appropriate for tree-like graphs [10], but not for other types of graphs. Ultrahyperbolic geometry has the advantage of generalizing both hyperbolic and spherical geometries and can describe relationships specific to those geometries. In particular, the geodesic *distance* can be written in the same way as the Poincaré and spherical distances as shown in Eq. (8), Some parts of the manifold are hyperbolic or spherical as explained in the paper. The converse is not true. The framework might then automatically learn representations to be part of a same hyperbolic or spherical part of the manifold depending on the context.

Those reasons led us to consider hierarchical graphs that were similar to trees, but where the presence of cycles in the graph limited the relevance of hyperbolic geometry. We experimentally validated our intuition. The choice of manifold of constant nonzero curvature (i.e. the optimal number of time and space dimensions $q + 1$ and $p$) then seems to depend on how much the graph is similar to a tree or to a graph where spherical geometry is appropriate, such as cycle graphs (or a mix of both).

It is also worth noting that ultrahyperbolic geometry can describe some graph concepts differently, if not better, than hyperbolic and spherical geometries. For instance, it is known that *triadic closure* is a concept in social network theory that is not valid for most large and complex networks. In other words, if $(\mathbf{x}, \mathbf{y})$ and $(\mathbf{x}, \mathbf{z})$ are strongly tied, triadic closure would imply that $(\mathbf{y}, \mathbf{z})$ are strongly tied as well. As explained in Section 3, the fact that we can find triplets that satisfy $d_\gamma(\mathbf{x}, \mathbf{y}) = d_\gamma(\mathbf{x}, \mathbf{z}) = 0$ but $d_\gamma(\mathbf{y}, \mathbf{z}) > 0$ allows to avoid triadic closure. Moreover, although we have applied our proposed representations to some type of graph, they can be applied to other applications that do not involve graphs. The possible applications are left for future research.

# C    Some properties about geodesics and logarithm map

## C.1    Geodesics of $\mathcal{Q}_\beta^{p,q}$

**Tangent space of pseudo-Riemannian submanifolds:** In the paper, we exploit the fact that our considered manifold $\mathcal{M}$ (here $\mathcal{Q}_\beta^{p,q}$) is a pseudo-Riemannian submanifold of $\overline{\mathcal{M}}$ (here $\mathbb{R}^{p,q+1}$). Since $\overline{\mathcal{M}}$ is chosen to be a vector space, we have a natural isomorphism between $\overline{\mathcal{M}}$ and its tangent space.

If $\mathcal{M}$ is a pseudo-Riemannian submanifold of $\overline{\mathcal{M}}$, we have the following direct sum decomposition:

$$T_{\mathbf{x}}(\overline{\mathcal{M}}) = T_{\mathbf{x}}(\mathcal{M}) + T_{\mathbf{x}}(\mathcal{M})^\perp, \tag{18}$$

where $T_{\mathbf{x}}(\mathcal{M})^\perp = \{\boldsymbol{\zeta} \in T_{\mathbf{x}}(\overline{\mathcal{M}}) : \forall \boldsymbol{\xi} \in T_{\mathbf{x}}(\mathcal{M}), \overline{g_{\mathbf{x}}}(\boldsymbol{\zeta}, \boldsymbol{\xi}) = 0\}$ is the orthogonal complement of $T_{\mathbf{x}}(\mathcal{M})$ and called the *normal space* of $\mathcal{M}$ at $\mathbf{x}$. It is a nondegenerate subspace of $T_{\mathbf{x}}(\overline{\mathcal{M}})$, and $\overline{g_{\mathbf{x}}}$ is

the metric at $\mathbf{x} \in \overline{\mathcal{M}}$. In the case of $\mathcal{Q}_\beta^{p,q}$, $T_\mathbf{x}(\mathcal{Q}_\beta^{p,q})^\perp$ is defined as:

$$T_\mathbf{x}(\mathcal{Q}_\beta^{p,q})^\perp = \{\boldsymbol{\zeta} \in T_\mathbf{x}\mathbb{R}^{p,q+1} : \forall \boldsymbol{\xi} \in T_\mathbf{x}\mathcal{Q}_\beta^{p,q}, \ \langle \boldsymbol{\zeta}, \boldsymbol{\xi} \rangle_q = 0\} \quad (19)$$

$$\approx N_\mathbf{x}(\mathcal{Q}_\beta^{p,q}, \mathbb{R}^{p,q+1}) = \{\lambda\mathbf{x} \in \mathbb{R}^{p,q+1} : \lambda \in \mathbb{R}\}. \quad (20)$$

where $\approx$ denotes isomorphism.

**Geodesic of a submanifold:** As mentioned in the main paper, a curve $\gamma$ is a geodesic if its acceleration is zero. However, the acceleration depends on the choice of the affine connection while the velocity does not (*i.e.* the velocity does not depend on the Christoffel symbols whereas the acceleration does, and different connections produce different geodesics, see details in page 66 of [21] or Chapter 5.4 of [1]). Let us note $\frac{\overline{\mathrm{D}}}{dt}\left(\gamma'(t)\right)$ (resp. $\frac{\mathrm{D}}{dt}\left(\gamma'(t)\right)$) the covariant derivative of $\gamma'(t)$ along $\gamma(t)$ in $\mathbb{R}^{p,q+1}$ (resp. in $\mathcal{Q}_\beta^{p,q}$). By using the *induced connection* (see page 98 of [21]) and the fact that $\mathcal{Q}_\beta^{p,q}$ is isometrically embedded in $\mathbb{R}^{p,q+1}$, the second-order ordinary differential equation about the zero acceleration of the geodesic is equivalent to (see page 103 of [21]):

$$\frac{\overline{\mathrm{D}}}{dt}\left(\gamma'(t)\right) \in N_{\gamma(t)}(\mathcal{Q}_\beta^{p,q}, \mathbb{R}^{p,q+1}) \iff \gamma''(t) = \frac{\mathrm{D}}{dt}\left(\gamma'(t)\right) = \Pi_{\gamma(t)}\left(\frac{\overline{\mathrm{D}}}{dt}\left(\gamma'(t)\right)\right) = \mathbf{0}. \quad (21)$$

where $\Pi_{\gamma(t)}$ is defined in Eq. (3) and orthogonally projects onto $T_{\gamma(t)}\mathcal{Q}_\beta^{p,q}$. In other words, $\gamma$ is straight in $\mathcal{M}$ but its curving in $\overline{\mathcal{M}}$ is the one forced by the curving of $\mathcal{M}$ itself in $\overline{\mathcal{M}}$ [21].

The initial conditions $\gamma_{\mathbf{x}\to\boldsymbol{\xi}}(0) = \mathbf{x} \in \mathcal{Q}_\beta^{p,q}$ and $\gamma'_{\mathbf{x}\to\boldsymbol{\xi}}(0) = \boldsymbol{\xi} \in T_\mathbf{x}\mathcal{Q}_\beta^{p,q}$ are straightforward. With the formulation of $\gamma_{\mathbf{x}\to\boldsymbol{\xi}}$ in Eq. (5), one can first verify that for all $t$, $\gamma_{\mathbf{x}\to\boldsymbol{\xi}}(t)$ lies on $\mathcal{Q}_\beta^{p,q}$. Indeed, since $\langle \mathbf{x}, \boldsymbol{\xi} \rangle_q = 0$, we have:

$$\forall t \in \mathbb{R}, \ \forall \mathbf{x} \in \mathcal{Q}_\beta^{p,q}, \ \forall \boldsymbol{\xi} \in T_\mathbf{x}\mathcal{Q}_\beta^{p,q}, \ \langle \gamma_{\mathbf{x}\to\boldsymbol{\xi}}(t), \gamma_{\mathbf{x}\to\boldsymbol{\xi}}(t) \rangle_q = \beta. \quad (22)$$

The acceleration of $\gamma$ at $t$ (in the ambient space $\mathbb{R}^{p,q+1}$) is in $N_{\gamma(t)}(\mathcal{Q}_\beta^{p,q}, \mathbb{R}^{p,q+1})$ since we have $\forall \mathbf{x} \in \mathcal{Q}_\beta^{p,q}, \ \forall \boldsymbol{\xi} \in T_\mathbf{x}\mathcal{Q}_\beta^{p,q}$:

$$\forall t \in \mathbb{R}, \ \frac{\overline{\mathrm{D}}}{dt}\left(\gamma'_{\mathbf{x}\to\boldsymbol{\xi}}(t)\right) = \frac{\langle \boldsymbol{\xi}, \boldsymbol{\xi} \rangle_q}{|\beta|}\gamma_{\mathbf{x}\to\boldsymbol{\xi}}(t) \in N_{\gamma_{\mathbf{x}\to\boldsymbol{\xi}}(t)}(\mathcal{Q}_\beta^{p,q}, \mathbb{R}^{p,q+1}) \quad (23)$$

$$\implies \forall t \in \mathbb{R}, \ \gamma''_{\mathbf{x}\to\boldsymbol{\xi}}(t) = \frac{\mathrm{D}}{dt}\left(\gamma'_{\mathbf{x}\to\boldsymbol{\xi}}(t)\right) = \mathbf{0}. \quad (24)$$

We have found a solution for our second-order ordinary differential equation, the solution is unique by definition. □

From its formulation in Eq. (5), the nonconstant geodesic $\gamma_{\mathbf{x}\to\boldsymbol{\xi}}$ is similar to the hyperbolic, flat or spherical case if $\langle \boldsymbol{\xi}, \boldsymbol{\xi} \rangle_q$ is positive, zero or negative, respectively.

## C.2 On the non existence of the logarithm map of some points

We explain here why $\log_\mathbf{x}(\mathbf{y})$ is not defined if $\langle \mathbf{x}, \mathbf{y} \rangle_q \geq |\beta|$. Our proof relies on the fact the domain of the exponential map is the whole tangent space. We also use the fact that $\forall \boldsymbol{\xi} \in T_\mathbf{x}\mathcal{Q}_\beta^{p,q}$, we have $\langle \mathbf{x}, \boldsymbol{\xi} \rangle_q = 0$ by definition of tangent vectors to $\mathcal{Q}_\beta^{p,q}$.

Assume that there exists a tangent vector $\boldsymbol{\xi} \in T_\mathbf{x}\mathcal{Q}_\beta^{p,q}$ such that $\mathbf{y} = \exp_\mathbf{x}(\boldsymbol{\xi})$ and $\langle \mathbf{y}, \mathbf{x} \rangle_q \geq |\beta|$. We consider the three possible cases of sign of $\langle \boldsymbol{\xi}, \boldsymbol{\xi} \rangle_q$.

• Let us first assume that $\langle \boldsymbol{\xi}, \boldsymbol{\xi} \rangle_q > 0$. For all $\boldsymbol{\xi} \in T_\mathbf{x}\mathcal{Q}_\beta^{p,q}$ that satisfies $\langle \boldsymbol{\xi}, \boldsymbol{\xi} \rangle_q > 0$, we have by definition of the corresponding exponential map:

$$\langle \mathbf{x}, \exp_\mathbf{x}(\boldsymbol{\xi}) \rangle_q = \|\mathbf{x}\|_q^2 \cosh\left(\frac{\sqrt{|\langle \boldsymbol{\xi}, \boldsymbol{\xi} \rangle_q|}}{\sqrt{|\beta|}}\right) + \sqrt{|\beta|}\frac{\langle \mathbf{x}, \boldsymbol{\xi} \rangle_q}{\sqrt{|\langle \boldsymbol{\xi}, \boldsymbol{\xi} \rangle_q|}}\sinh\left(\frac{\sqrt{|\langle \boldsymbol{\xi}, \boldsymbol{\xi} \rangle_q|}}{\sqrt{|\beta|}}\right) \quad (25)$$

$$= \|\mathbf{x}\|_q^2 \cosh\left(\frac{\sqrt{|\langle \boldsymbol{\xi}, \boldsymbol{\xi} \rangle_q|}}{\sqrt{|\beta|}}\right) < 0. \quad (26)$$

Therefore, there exists no tangent vector $\boldsymbol{\xi} \in T_{\mathbf{x}}\mathcal{Q}_{\beta}^{p,q}$ that satisfies $\langle \boldsymbol{\xi}, \boldsymbol{\xi} \rangle_q > 0$ and $\langle \mathbf{x}, \exp_{\mathbf{x}}(\boldsymbol{\xi}) \rangle_q > |\beta| > 0$.

• Let us now assume that $\langle \boldsymbol{\xi}, \boldsymbol{\xi} \rangle_q = 0$. For all $\boldsymbol{\xi} \in T_{\mathbf{x}}\mathcal{Q}_{\beta}^{p,q}$ that satisfies $\langle \boldsymbol{\xi}, \boldsymbol{\xi} \rangle_q = 0$, we have:

$$\langle \mathbf{x}, \exp_{\mathbf{x}}(\boldsymbol{\xi}) \rangle_q = \langle \mathbf{x}, \mathbf{x} + \boldsymbol{\xi} \rangle_q = \langle \mathbf{x}, \mathbf{x} \rangle_q + \langle \mathbf{x}, \boldsymbol{\xi} \rangle_q = \langle \mathbf{x}, \mathbf{x} \rangle_q = \beta < 0. \tag{27}$$

• Let us now assume that $\langle \boldsymbol{\xi}, \boldsymbol{\xi} \rangle_q < 0$. For all $\boldsymbol{\xi} \in T_{\mathbf{x}}\mathcal{Q}_{\beta}^{p,q}$ that satisfies $\langle \boldsymbol{\xi}, \boldsymbol{\xi} \rangle_q < 0$, we have:

$$\langle \mathbf{x}, \exp_{\mathbf{x}}(\boldsymbol{\xi}) \rangle_q = \|\mathbf{x}\|_q^2 \cos\left( \frac{\sqrt{|\langle \boldsymbol{\xi}, \boldsymbol{\xi} \rangle_q|}}{\sqrt{|\beta|}} \right) + \sqrt{|\beta|}\frac{\langle \mathbf{x}, \boldsymbol{\xi} \rangle_q}{\sqrt{|\langle \boldsymbol{\xi}, \boldsymbol{\xi} \rangle_q|}} \sin\left( \frac{\sqrt{|\langle \boldsymbol{\xi}, \boldsymbol{\xi} \rangle_q|}}{\sqrt{|\beta|}} \right) \tag{28}$$

$$= \|\mathbf{x}\|_q^2 \cos\left( \frac{\sqrt{|\langle \boldsymbol{\xi}, \boldsymbol{\xi} \rangle_q|}}{\sqrt{|\beta|}} \right) \in [\beta, |\beta|]. \tag{29}$$

Given the formulation of the geodesic in Eq. (5), $\langle \mathbf{x}, \mathbf{y} \rangle_q = \langle \mathbf{x}, \exp_{\mathbf{x}}(\boldsymbol{\xi}) \rangle_q \geq |\beta|$ iff $\mathbf{y} = -\mathbf{x}$.

From our study above, there exists no geodesic (hence no logarithm map) joining $\mathbf{x}$ and $\mathbf{y}$ if $\langle \mathbf{x}, \mathbf{y} \rangle_q \geq |\beta|$ except if $\mathbf{y} = -\mathbf{x}$. The antipodal point $\mathbf{y} = -\mathbf{x}$ is also a special case for which there does not exist a logarithm map since $\mathbf{x}$ and $-\mathbf{x}$ are joined by infinitely many minimizing geodesics of equal length (a similar case is the $q$-sphere). Nonetheless, the geodesic "distance" between $\mathbf{x}$ and $-\mathbf{x}$ can be defined as $\pi\sqrt{|\beta|}$. □

## C.3 Anti-isometry between $\mathcal{Q}_{\beta}^{p,q}$ and $\mathcal{Q}_{-\beta}^{q+1,p-1}$

In the main paper, we state that there is an anti-isometry between $\mathcal{Q}_{\beta}^{p,q}$ and $\mathcal{Q}_{-\beta}^{q+1,p-1}$. We recall that $\mathcal{Q}_{-\beta}^{q+1,p-1}$ is embedded in $\mathbb{R}^{q+1,p}$. Let us note $\sigma : \mathbb{R}^{p,q+1} \to \mathbb{R}^{q+1,p}$ the mapping defined as:

$$\forall \mathbf{x} = (x_0, \cdots, x_{p+q})^\top \in \mathbb{R}^{p,q+1}, \ \sigma(\mathbf{x}) = (x_{p+q}, x_{p+q-1}, \cdots, x_1, x_0)^\top. \tag{30}$$

We have:
$$\forall \mathbf{x} \in \mathbb{R}^{p,q+1}, \mathbf{y} \in \mathbb{R}^{p,q+1}, \ \langle \mathbf{x}, \mathbf{y} \rangle_q = -\langle \sigma(\mathbf{x}), \sigma(\mathbf{y}) \rangle_{p-1}. \tag{31}$$

## C.4 Curvature of $\mathcal{Q}_{\beta}^{p,q}$

The manifold $\mathcal{Q}_{\beta}^{p,q}$ is a total umbilic hypersurface of $\mathbb{R}^{p,q+1}$ and has constant sectional curvature $1/\beta$ and constant mean curvature $\kappa = |\beta|^{-1/2}$ with respect to the unit normal vector field $\mathcal{N}(\mathbf{x}) = -\kappa\mathbf{x}$. More details can be found in Chapter 3 of [2].

## C.5 Proof of Theorem 4.1

We give the proofs for Theorem 4.1 that we recall below:

**Theorem C.1** (Diffeomorphisms). *For any $\beta < 0$, there is a diffeomorphism $\psi : \mathcal{Q}_{\beta}^{p,q} \to \mathcal{S}^q \times \mathbb{R}^p$. Let us note $\mathbf{x} = \begin{pmatrix} \mathbf{t} \\ \mathbf{s} \end{pmatrix} \in \mathcal{Q}_{\beta}^{p,q}$ with $\mathbf{t} \in \mathbb{R}_*^{q+1}$ and $\mathbf{s} \in \mathbb{R}^p$, let us note $\mathbf{z} = \begin{pmatrix} \mathbf{u} \\ \mathbf{v} \end{pmatrix} \in \mathcal{S}^q \times \mathbb{R}^p$ where $\mathbf{u} \in \mathcal{S}^q$ and $\mathbf{v} \in \mathbb{R}^p$. The mapping $\psi$ and its inverse $\psi^{-1}$ are formulated:*

$$\psi(\mathbf{x}) = \begin{pmatrix} \frac{1}{\|\mathbf{t}\|}\mathbf{t} \\ \frac{1}{\sqrt{|\beta|}}\mathbf{s} \end{pmatrix} \qquad and \qquad \psi^{-1}(\mathbf{z}) = \sqrt{|\beta|}\begin{pmatrix} \sqrt{1 + \|\mathbf{v}\|^2}\mathbf{u} \\ \mathbf{v} \end{pmatrix}. \tag{32}$$

We need to show that $\psi(\psi^{-1}(\mathbf{z})) = \mathbf{z}$ and $\psi^{-1}(\psi(\mathbf{x})) = \mathbf{x}$.

**Space dimensions:** The mapping of the space dimensions of $\mathbf{x}$ to the space dimensions of $\mathbf{z}$ simply involves a scaling factor of $\sqrt{|\beta|}$ (*i.e.* $\mathbf{v} = \frac{1}{\sqrt{|\beta|}}\mathbf{s}$, and $\mathbf{s} = \sqrt{|\beta|}\mathbf{v}$).

**Time dimensions:** We show the invertibility of the mappings taking time dimensions as input.

• Let us first show $\psi^{-1}(\psi(\mathbf{x})) = \mathbf{x}$. We recall that $\mathbf{x} = \begin{pmatrix} \mathbf{t} \\ \mathbf{s} \end{pmatrix} \in \mathcal{Q}_{\beta}^{p,q}$ with $\mathbf{t} \in \mathbb{R}_*^{q+1}$ and $\mathbf{s} \in \mathbb{R}^p$.

We need to show that:

$$\mathbf{t} = \sqrt{|\beta|}\sqrt{1 + \|\mathbf{v}\|^2}\frac{\mathbf{t}}{\|\mathbf{t}\|} = \sqrt{|\beta|}\sqrt{1 + \|\frac{1}{\sqrt{|\beta|}}\mathbf{s}\|^2}\frac{\mathbf{t}}{\|\mathbf{t}\|} = \sqrt{|\beta|}\sqrt{1 + \frac{1}{|\beta|}\|\mathbf{s}\|^2}\frac{\mathbf{t}}{\|\mathbf{t}\|}. \tag{33}$$

To show Eq. (33), it is sufficient to prove that the following property is satisfied:

$$\|\mathbf{t}\| = \sqrt{|\beta|}\sqrt{1 + \frac{1}{|\beta|}\|\mathbf{s}\|^2}, \text{which is satisfied if } \|\mathbf{t}\|^2 = |\beta|(1 + \frac{1}{|\beta|}\|\mathbf{s}\|^2) = |\beta| + \|\mathbf{s}\|^2. \tag{34}$$

Since $\mathbf{x} \in \mathcal{Q}_\beta^{p,q}$, we have by definition $\|\mathbf{x}\|_q^2 = \|\mathbf{s}\|^2 - \|\mathbf{t}\|^2 = \beta < 0$. Therefore, we have:

$$\|\mathbf{t}\|^2 = \|\mathbf{s}\|^2 - \beta = \|\mathbf{s}\|^2 + |\beta|. \tag{35}$$

• Let us now show $\psi(\psi^{-1}(\mathbf{z})) = \mathbf{z}$. We recall that $\mathbf{z} = \begin{pmatrix} \mathbf{u} \\ \mathbf{v} \end{pmatrix} \in \mathcal{S}^q \times \mathbb{R}^p$ where $\mathbf{u} \in \mathcal{S}^q$ and $\mathbf{s} \in \mathbb{R}^p$.

We need to show:

$$\mathbf{u} = \frac{\sqrt{|\beta|}\sqrt{1 + \|\mathbf{v}\|^2}\mathbf{u}}{\|\sqrt{|\beta|}\sqrt{1 + \|\mathbf{v}\|^2}\mathbf{u}\|} = \frac{\mathbf{u}}{\|\mathbf{u}\|}. \tag{36}$$

By definition of $\mathcal{S}^q$, $\mathbf{u}$ satisfies Eq. (36). $\qquad\qquad\qquad\qquad\qquad\qquad\qquad\qquad\qquad\qquad\qquad\square$

## C.6   The metric tensor is positive definite only if $q = 0$ and indefinite in the general case

We show here that the hyperboloid (*i.e.* upper sheet of the two-sheet hyperboloid $\mathcal{Q}_\beta^{p,0}$ with $p \geq 1$) is a Riemannian manifold even if its symmetric metric matrix $\mathbf{G} = \mathbf{I}_{1,p}$ is not positive definite. That result is already known in the literature and a proof can be found in [23]. We still give it because we think it is important.

We recall that a Riemannian manifold is a pseudo-Riemannian manifold with positive definite metric tensor. In other words, for all $\mathbf{x} \in \mathcal{M}$ where $\mathcal{M}$ is a pseudo-Riemannian manifold, $\mathcal{M}$ is Riemannian iff $\forall \boldsymbol{\xi} \in T_\mathbf{x}\mathcal{M}$, $g_\mathbf{x}(\boldsymbol{\xi}, \boldsymbol{\xi}) > 0$ iff $\boldsymbol{\xi} \neq 0$.

We recall that we consider that $\beta < 0$, the usual definition of the hyperboloid $\mathcal{H}_\beta^p$ in the literature is:

$$\mathcal{H}_\beta^p = \left\{\mathbf{x} = (x_0, x_1, \cdots, x_p)^\top \in \mathcal{Q}_\beta^{p,0} : x_0 \geq 0\right\}, \tag{37}$$

Let us note $\mathbf{x} = (x_0, x_1, \cdots, x_p)^\top \in \mathcal{H}_\beta^p$ and $\boldsymbol{\xi} = (\xi_0, \cdots, \xi_p)^\top$ a tangent vector at $\mathbf{x}$. For simplicity, we will denote the vectors $\mathbf{u} = (x_1, \cdots, x_p)^\top \in \mathbb{R}^p$ and $\mathbf{v} = (\xi_1, \cdots, \xi_p)^\top \in \mathbb{R}^p$ so that $\mathbf{x} = \begin{pmatrix} x_0 \\ \mathbf{u} \end{pmatrix} \in \mathcal{H}_\beta^p$ and $\boldsymbol{\xi} = \begin{pmatrix} \xi_0 \\ \mathbf{v} \end{pmatrix}$. By definition of $\mathcal{H}_\beta^p$, we have $x_0 = \sqrt{-\beta + \|\mathbf{u}\|^2} > 0$.

Moreover, by definition of the tangent space, and since $\mathcal{H}_\beta^p$ is a subset of $\mathcal{Q}_\beta^{p,0}$, any tangent vector $\boldsymbol{\xi}$ satisfies $0 = \langle \mathbf{x}, \boldsymbol{\xi} \rangle_q = -x_0\xi_0 + \langle \mathbf{u}, \mathbf{v} \rangle$ where $q = 0$. This implies:

$$\xi_0 = \frac{\langle \mathbf{u}, \mathbf{v} \rangle}{x_0} = \frac{\langle \mathbf{u}, \mathbf{v} \rangle}{\sqrt{-\beta + \|\mathbf{u}\|^2}} \implies \xi_0^2 = \frac{\langle \mathbf{u}, \mathbf{v} \rangle^2}{-\beta + \|\mathbf{u}\|^2}. \tag{38}$$

It is obvious that if $\boldsymbol{\xi} = \mathbf{0}$, then $\langle \boldsymbol{\xi}, \boldsymbol{\xi} \rangle_q = 0$. To show that $\mathcal{H}_\beta^p$ is a Riemannian manifold, we need to show that every non-vanishing tangent vector $\boldsymbol{\xi}$ satisfies $\langle \boldsymbol{\xi}, \boldsymbol{\xi} \rangle_q > 0$. By definition of $\langle \cdot, \cdot \rangle_q$, we have

$$\langle \boldsymbol{\xi}, \boldsymbol{\xi} \rangle_q = -\xi_0^2 + \|\mathbf{v}\|^2. \tag{39}$$

We then need to show that $\xi_0^2 < \|\mathbf{v}\|^2$. By the Cauchy-Schwarz inequality, we have $\langle \mathbf{u}, \mathbf{v} \rangle^2 \leq \|\mathbf{v}\|^2\|\mathbf{u}\|^2 \leq \|\mathbf{v}\|^2(-\beta + \|\mathbf{u}\|^2)$. This implies from Eq. (38):

$$\xi_0^2 = \frac{\langle \mathbf{u}, \mathbf{v} \rangle^2}{-\beta + \|\mathbf{u}\|^2} \leq \|\mathbf{v}\|^2. \tag{40}$$

By using the Cauchy-Schwarz inequality, Eq. (40) can be an equality iff $\boldsymbol{\xi} = \mathbf{0}$ since $\|\mathbf{u}\|^2 < -\beta + \|\mathbf{u}\|^2$. Since every non-vanishing tangent vector $\boldsymbol{\xi}$ to $\mathcal{H}_\beta^p$ satisfies $\langle \boldsymbol{\xi}, \boldsymbol{\xi} \rangle_q > 0$, the metric tensor of $\mathcal{H}_\beta^p$ is positive definite. This shows that $\mathcal{H}_\beta^p$ is Riemannian. $\qquad\square$

The fact that we consider $\beta < 0$ is important. For instance, if $\beta = 1$, then let us note $\mathbf{x} = (0, 1, 0)^\top \in \mathcal{Q}_1^{2,0}$. If $\boldsymbol{\xi} = (1, 0, 0)^\top \in T_\mathbf{x} \mathcal{Q}_1^{2,0}$, then we have $\langle \boldsymbol{\xi}, \boldsymbol{\xi} \rangle_q < 0$. If $\boldsymbol{\xi} = (0, 0, 1)^\top \in T_\mathbf{x} \mathcal{Q}_1^{2,0}$, then we have $\langle \boldsymbol{\xi}, \boldsymbol{\xi} \rangle_q > 0$. Finally, if $\boldsymbol{\xi} = (1, 0, 1)^\top \in T_\mathbf{x} \mathcal{Q}_1^{2,0}$, then we have $\langle \boldsymbol{\xi}, \boldsymbol{\xi} \rangle_q = 0$. The metric tensor is then indefinite if $\beta = 1$, and $\mathcal{Q}_1^{2,0}$ is non-Riemannian. That result can be extended to any positive value of $\beta$ by considering instead $\mathbf{x} = (0, \sqrt{\beta}, 0)^\top$.

**Indefiniteness:** If $\beta < 0$, one can show in a similar way that $\langle \cdot, \cdot \rangle_q$ is indefinite if $q \geq 1$ and $p \geq 1$. Let us consider $q \geq 1, p \geq 1$ and $\mathbf{x} = (0, \sqrt{|\beta|}, \cdots, 0)^\top \in \mathcal{Q}_\beta^{p,q}$ where only the second element of $\mathbf{x}$ is nonzero. If $\boldsymbol{\xi} \in T_\mathbf{x} \mathcal{Q}_\beta^{p,q}$ is defined such that all its elements except the first one are equal to zero, then $\langle \boldsymbol{\xi}, \boldsymbol{\xi} \rangle_q < 0$. If $\boldsymbol{\xi} \in T_\mathbf{x} \mathcal{Q}_\beta^{p,q}$ is defined such that all its elements except the last one are equal to zero, then $\langle \boldsymbol{\xi}, \boldsymbol{\xi} \rangle_q > 0$. The scalar product $\langle \cdot, \cdot \rangle_q$ is then indefinite if $q \geq 1$ and $p \geq 1$.

# D   Experimental Results

We complete here the experimental result section. First, we provide additional details about Zachary's karate club dataset experiments in Section D.1. We provide in Section D.2 similar experimental results on a larger social network dataset about co-authorship at NIPS conferences.

## D.1   Additional details about Zachary's karate club experiments

**Complexity:** We implemented our approach in PyTorch 1.0 [22] which automatically calculates the Euclidean gradient $\nabla f(\mathbf{x}_i)$ for each node $v_i \in V$. Once $\nabla f(\mathbf{x}_i)$ is computed, the computation of $\boldsymbol{\chi}_i = \Pi_\mathbf{x}(\mathbf{G}\Pi_\mathbf{x}(\mathbf{G}\nabla f(\mathbf{x}_i)))$ is linear in the dimensionality $d$ and is very efficient to compute. The exponential map is also efficient to calculate (*i.e.* linear algorithmic complexity).

**Running time:** The step size/learning rate for all our optimizers is fixed to $\eta = 10^{-6}$ (without momentum). Step size strategies could have been used to improve convergence rate but the goal of our experiment was only to verify that our solvers decrease the optimized function.

In the weighted case, we stop after 10,000 iterations of descent method. The training process takes 182 seconds ($\sim$3 minutes) on an Intel i7-7700k CPU, and 254 seconds ($\sim$4 minutes) on an Intel i7-8650U CPU when using the pseudo-Riemannian optimizer introduced in Section 4.2. The optimizer introduced in Section 4.1 is 10% faster (165 seconds on an Intel i7-7700k CPU) since it requires less computations.

## D.2   Co-authorship from papers published at NIPS

We also quantitatively evaluate our approach on a dataset that describes co-authorship information from papers published at NIPS from 1988 to 2003 [9].

**Description of the dataset:** The graph $G = (V, E)$ is constructed by considering each author as a node, and an edge $e_k = (v_i, v_j)$ is created iff the authors $i$ and $j$ are co-authors of at least one paper. The capacity $c_k$ is the number of papers co-authored by the pair of authors $e_k = (v_i, v_j)$. The original number of authors in the dataset is 2865, but only $n = |V| = 2715$ authors have at least one co-author. The number of edges is $m = |E| = 4733$, which means that the number of pairs of nodes that have no edge joining them is 3,679,522. The capacity $c_k$ of each edge $e_k$ is a natural number in $\{1, \cdots, 9\}$. The "leaders" of this dataset are not the authors with highest number of papers, but those with highest number of co-authors.

**Implementation details and running times:** We ran all our experiments for this dataset on a 12 GB NVIDIA TITAN V GPU. To fit into memory, when constructing each weaker set $\mathcal{W}(e_k)$, we take into account all the edges with capacity lower than $c_k$ and randomly sample 42,000 pairs of nodes without edge joining them. Our fixed hyperparameters are: $\beta = -1$, the temperature is $\tau = 10^{-5}$, and the step size is $\eta = 10^{-8}$. We stop the training of representations lying on 4-dimensional manifolds (see Table 2) after 10,000 iterations, which takes 10 hours to train ($\sim 1,000$ iterations per hour). We stop

Table 2: Evaluation scores for the different learned representations lying on 4-dimensional manifolds on the NIPS dataset

| Evaluation metric | $\mathbb{R}^4$ (flat) | $\mathcal{Q}_{-1}^{4,0}$ (hyperbolic) | $\mathcal{Q}_{-1}^{3,1}$ (ours) | $\mathcal{Q}_{-1}^{2,2}$ (ours) | $\mathcal{Q}_{-1}^{1,3}$ (ours) | $\mathcal{Q}_{-1}^{0,4}$ (spherical) |
|---|---|---|---|---|---|---|
| Spearman's $\rho$ for the whole dataset | 0.469 | 0.460 | 0.629 | **0.667** | 0.625 | 0.437 |
| Spearman's $\rho$ for $s_i \geq 10$ | 0.512 | 0.490 | **0.552** | 0.493 | 0.441 | 0.493 |
| Spearman's $\rho$ for $s_i \geq 20$ | 0.217 | 0.292 | 0.316 | 0.307 | 0.227 | **0.387** |
| Recall@1 (in %) | 70.7 | 70.2 | **83.5** | 82.6 | 78.1 | 69.7 |

Table 3: Evaluation scores for the different learned representations lying on 6-dimensional manifolds on the NIPS dataset

| Evaluation metric | $\mathbb{R}^6$ (flat) | $\mathcal{Q}_{-1}^{6,0}$ (hyperbolic) | $\mathcal{Q}_{-1}^{5,1}$ (ours) | $\mathcal{Q}_{-1}^{4,2}$ (ours) | $\mathcal{Q}_{-1}^{3,3}$ (ours) | $\mathcal{Q}_{-1}^{2,4}$ (ours) | $\mathcal{Q}_{-1}^{1,5}$ (ours) | $\mathcal{Q}_{-1}^{0,6}$ (spherical) |
|---|---|---|---|---|---|---|---|---|
| Spearman's $\rho$ for the whole dataset | 0.554 | 0.455 | 0.617 | 0.666 | **0.688** | 0.675 | 0.610 | 0.456 |
| Spearman's $\rho$ for $s_i \geq 10$ | 0.459 | 0.515 | 0.507 | 0.501 | 0.514 | 0.470 | 0.494 | **0.532** |
| Spearman's $\rho$ for $s_i \geq 20$ | 0.372 | 0.321 | 0.211 | 0.317 | **0.383** | 0.370 | 0.213 | 0.257 |
| Recall@1 (in %) | 88.6 | **99.9** | 93.1 | 96.4 | 96.7 | 95.7 | 93.5 | **99.9** |

the training of 6-dimensional manifolds (see Table 3) after 25,000 iterations because they take longer to converge.

**Quantitative evaluation:** The evaluation process is the same as the evaluation for Zachary's karate club dataset. However, unlike for Zachary's karate club dataset, the factions and their leaders are unknown. We then only consider Spearman's rank correlation coefficient scores. We also consider the average recall at 1 as explained below.

The capacity matrix $\mathbf{C} \in \{0, 1, \cdots, 8, 9\}^{n \times n}$ is symmetric, so our score matrix is $\mathbf{S} = \mathbf{C}$. We select the most influential members based on the score $s_i = \sum_{j=1}^{n} \mathbf{S}_{ij} \in \{1, \cdots, 97\}$. Spearman's rank correlation coefficient [24] between the selected ordered scores $s_i$ and corresponding $\delta_i = \sum_{j=1}^{n} \mathsf{d}(\mathbf{x}_i, \mathbf{x}_j)$ is reported in Table 2 and shows the relevance of our representations. The most influential members are selected if their score $s_i$ is at least 1 (*i.e.* top $n$ members where $n = 2715$ is the number of nodes), at least 10 (*i.e.* top 232 authors) and at least 20 (*i.e.* top 57 authors). We also report the average recall at 1 (in percentage): for each node $v_i$, we find the nearest neighbor $v_j$ *w.r.t.* the chosen metric, the recall at 1 is 0 if $\mathbf{C}_{ij} = 0$, and 1 otherwise. Both $Q_{-1}^{3,1}$ $Q_{-1}^{2,2}$ outperform the hyperbolic model.

If we use the Euclidean norm of hyperbolic representations (in $\mathcal{Q}_{-1}^{4,0}$) as proxy, we obtain the following scores: 0.044, -0.113, 0.264 for the top 2715, to 232 and top 57 members respectively. These scores are worse than using $\delta_i$ as proxy due to the presence of cycles in the graph.

In the 6-dimensional manifold case, the Recall@1 score is better for the hyperbolic and spherical representations. On the other hand, the ultrahyperbolic representations perform better in terms of Spearman's rank correlation coefficient.

In conclusion, our proposed ultrahyperbolic representations extract hierarchy better than hyperbolic representations when the hierarchy graph contains cycles.