[Reviews · NeurIPS 2020]

Review 1

Summary and Contributions: The paper presents a generalization of the hyperbolic and spherical manifolds. The main contribution of the paper is to show the computation of geodesics, exponential, and logarithm maps that allow optimization on this manifold. The experiment results look solid.

Strengths: A good work on a particular manifold, especially for optimization on the manifold. It details relevant developments.

Weaknesses: The paper could benefit from a few more explanations as mentioned in the detailed feedback.

Correctness: Yes. The developments look correct though I have not done a deep-dive over all the proofs.

Clarity: Yes, mostly. Certain parts can be improved (see comments in detailed feedback)

Relation to Prior Work: Yes.

Reproducibility: Yes

Additional Feedback: While the paper is mostly easy to follow, it becomes confusing at certain places. The metric discussion is missing (at least not highlighted properly), e.g., the definition of g_x(.,.) I understand that it means <>_q in (1), but it should be explicitly called out. Discuss the non positive definiteness of the metric on the tangent space explicitly. Without loss of generality, \beta can be fixed to -1. This will make certain expressions fewer notations heavy. In Section 3, the discussion between lines 83 to 88 looks odd and not relevant. Where is (4) used? It would be good to have a table in the paper that shows all the manifold related formulas in one place for easy referencing. While Figure 3 is great, it would be great to plot loss versus time to show the benefit of the proposed optimization approach over the Euclidean modeling. It would be great to have the discussion in Lines 496 to 500 in the main paper as well. ------- After rebuttal -------- I appreciate the author's response.


Review 2

Summary and Contributions: Authors propose a new representation for structured data: the ultrahyperbolic representation. It combines spherical and hyperbolic geometries by "stacking" them. Several expressions are given for this space, including geodesics, expo/log maps and a dissimilarity function. Optimization frameworks are then proposed to optimise a given cost function in that space. The optimisation deserves a specific attention as tangent vectors can have negative square norms, and authors propose a novel direction scheme. Experiments on the karate club dataset (+nips dataset on the supplementary) are then given. AFTER REBUTAL ------------------- I thank the authors for their response. I still think that a deeper investigation of the conditions/cases on which the geometry can be useful is missing, and after reding the rebutal, my concerns on this part remain. Nevertheless, I believe that the theoretical developments of the paper are interesting and that the paper may be published at NeurIPS.

Strengths: The paper is theoretical and introduces the ultrahyperbolic geometry, that combines both spherical and hyperbolic geometries. Tools to optimise on this geometry are then given, paving the way for new applications in machine learning. The experiments on the karate dataset show that the geometry can be useful when graphs contains cycles, which is a typical scenario on which the hyperbolic geometry fails (as it is supposed to represent tree-like data). As such, the work is a pioneer work in the field of hyperbolic geometry for machine learning ; the claims are sounds.

Weaknesses: The main weaknesses of the paper are the following: - it is unclear in which case this geometry can be useful "in real applications". In the experiments, authors state that it better embeds graph that contain cycles: a deeper analysis of this statement should be provided, together with all other relevant cases. If I have a problem with some tree- or graph-like data, in which case should I consider the ultrahyperbolic geometry rather than the hyperbolic one? - regarding the experimental evaluation, several questions are also raised. Authors propose to use the capacity of the nodes to learn the representations, as it is difficult to get ancestors in the presence of cycles. How does the ultrahyperbolic geometry (with q>1) behaves in "classical" scenario in which the hyperbolic assumption makes sense? How choosing the appropriate q value ? To what phenomenom it relates to?

Correctness: To the best of my knowledge, the claims are correct.

Clarity: The paper is well written and easy to follow.

Relation to Prior Work: yes

Reproducibility: Yes

Additional Feedback: It is probably beyond the scope of the paper but I was wondering how dedicated methods for graphs embedding behave w.r.t. the proposed method in the karateclub dataset?


Review 3

Summary and Contributions: The paper considers a pseudo-Riemannian geometry that extend the traditional hyperbolic and spherical space. Some theory regarding the geometry and optimization is provided. An application on graph-valued data is provided. == After the rebuttal == I have lowered my score for this paper after the rebuttal. My main concern with the paper is that the theory is too decoupled from the application. I cannot tell a reason why the developed representation should be applicable to graph-data. I had given the authors the benefit of the doubt hoping for an explanation in the rebuttal, but the rebuttal did not have a (to this reviewer) convincing argument why the representation should work well for graphs.

Strengths: The key contribution of the paper is theoretical. As far as I can tell, the paper provides two novel theoretical results: *) Closed-form expression for geodesics in the considered space is provided. Derivations seem to closely match corresponding results for spheres and hyperbolic space. This is to be expected; and the result remain novel. *) From my perspective the most interesting result is the derivation of a non-trivial descent direction over the considered space.

Weaknesses: The key contribution of the present paper is theoretical. The main issue is that the theory seems largely to have limited practical use. The paper propose to use the considered space for representing graphs, but this appears to largely be ad hoc with little to no grounding in first principles. The argument is that since trees are known to be well-represented in hyperbolic spaces, then graphs should naturally fit into a space that extend hyperbolic space. From what I can tell, this conclusion has no grounding in mathematics. If I have missed something, I would appreciate if the authors provide more detail in the rebuttal.

Correctness: The theory appear to be correct. The empirical work appears to be largely heuristic.

Clarity: The paper is generally easy to read, but may require prior experience in differential geometry. Given the theoretical nature of the paper, I would have preferred that a larger portion of the paper was devoted to explaining and deriving the theory (the current paper largely states results and refer the reader the supplementary material for details).

Relation to Prior Work: Yes. I miss a discussion of the work of Feragen et al. that were among the first to give tree-data a well-defined geometry: (*) Means in spaces of tree-like shapes A Feragen, S Hauberg, M Nielsen, F Lauze 2011 International Conference on Computer Vision (*) Towards a theory of statistical tree-shape analysis A Feragen, P Lo, M de Bruijne, M Nielsen, F Lauze IEEE Transactions on Pattern Analysis and Machine Intelligence

Reproducibility: Yes

Additional Feedback: Generally, I would wish for a greater emphasis on explaining and deriving the theory as this is where the main contributions lie. I had trouble understanding Eq. 9. In particular, I did not understand the motivation for the 'otherwise' branch -- where does this expression come from? In general, I found the graph-analysis aspect of the paper less convincing and this is the reason for my fairly low score. It seems to me that quite a few 'hacks' are needed to force graphs into the proposed representation (lines 220-229; the symmetrization of C in line 241). If I an missing a great principle here, then I would be interested to learn more.


Review 4

Summary and Contributions: This paper proposed a novel approach in representation learning context using by studying Riemannian geometry properties of pseudo-hyperboloids. The novelty of this paper is mainly theoretical. In particular, the authors proposed for the first time the explicit formulas of the geodesic distance on ultrahyperbolic manifolds and the logarithm map for pseudo-hyperboloids. Also, an optimization scheme using the ultrahyperbolic geometry is proposed, in which the authors have shown a relationship to representation learning of non-tree graphs both theoretically and experimentally.

Strengths: The theoretical grounding of this paper is solid. With precise geometric analysis, the authors further develop theories on pseudo-hyperboloids, e.g. Riemannian metrics, geodesics, exponential maps, and (pseudo-)distances. Formulas are explicit and concrete, and thus laid a solid foundation for numerical computations. For the numerical algorithm part, the author proposed a novel approach of computing representation learning, with the aim of improving performance on non-tree graphs. The experimental results also the efficacy of the proposed algorithm and its performance on various datasets.

Weaknesses: Theoretically, the dissimilarity metric (Eq.8) is locally defined due to the (non)existence of the logarithm map in the global scope. The effectiveness of the dissimilar metric is limited by the geometry of the underlying manifold, which might in turn limit the effectiveness of the proposed method on complicated real-world applications. The authors could have justified the applicability of the algorithm by experimenting it on more real-world datasets, alongside the two reported in the paper and the supplementary materials. The proposed optimization algorithm, as the authors pointed out in the conclusion section, lacks a convergence rate analysis. However, this is not a major issue as this is not the emphasis of the paper. Finally, it would be great if more experiments were conducted with various real-world datasets and under different settings of model hyper-parameters.

Correctness: This paper involves intensive Riemannian geometry computations, which require extensive background knowledge. However, the statements are clear and well organized. And the empirical methodology is consistent with previous state-of-arts.

Clarity: The paper is well written, all the mathematical concepts are explained clearly. The computational algorithms are represented in details.

Relation to Prior Work: This paper proposed a novel method based on the geometries of pseudo-hyperboloids, which generalized the hyperboloid models studied in existing representation learning literature. Both of the theoretic and numerical novelties are clearly stated in the paper.

Reproducibility: Yes

Additional Feedback: The paper is well presented and novelties are clearly stated, and paved a new way of solving representation leaning problems with non-Riemannian nature. AFTER REBUTAL ------------------- I thank the authors for their response. My concern has been fully addressed, especially why the proposed method is better than hyperbolic geometry for graph embedding/representation. I believe that the theoretical developments of the paper are interesting and that the paper should be published at NeurIPS. So I keep my original score.

[Author Response · NeurIPS 2020]

We thank the reviewers for their positive and valuable feedback. We recall that our paper proposes a general framework to learn **ultrahyperbolic** representations. The proposed representations lie on a pseudo-Riemannian manifold with constant nonzero curvature, they generalize both hyperbolic and spherical representations that are popular in machine learning. The main difficulty of learning such representations is that they lie on a manifold whose metric need not be positive definite, and the manifold is non-Riemannian in most cases (except for the hyperbolic and spherical cases as explained in the paper). We introduce the necessary differential geometry tools (e.g. geodesics, exponential/logarithm maps) to measure dissimilarity between points, and also propose optimizers for differentiable functions defined on such manifolds. In particular, we explain why the pseudo-Riemannian gradient is not a descent direction. We then propose a simple, efficient and non-trivial descent direction defined in the tangent space (see Eq. (12)).

**Improving readability:** Our contributions are mainly theoretical, and we agree with most reviewers (**R1**,**R3**) that the pseudo-Riemannian optimizer introduced in Section 4.2 is a major contribution. Due to lack of space, we provided the detailed explanations with proofs in the supp. material. However, according to the NeurIPS 2020 website, camera-ready versions are allowed a ninth content page. To improve readability, we will include the extended version of the optimizer subsection in the main paper, if accepted. We will also account for the suggestions of the reviewers as follows.

**R1**: Thank you for your suggestions. **(1)** We will indicate in Section 2 that for any $\beta < 0$, $\mathcal{Q}_\beta^{p,q}$ is homothetic to $\mathcal{Q}_{-1}^{p,q}$, $\beta$ can then be considered to be $-1$. **(2)** We did not exploit the extrinsic distance in Eq. (4), except in the null geodesic case since the formulation is similar in this particular case. The goal of lines 83-88 was to explain that many machine learning approaches consider the extrinsic geometry (i.e. ambient space distance) of the spherical or hyperbolic manifold, or its intrinsic geometry (i.e. geodesic distance). Since both distances are increasing functions of each other in the Riemannian cases, choosing one or the other has no major impact. This is not the case in the ultrahyperbolic case, which is why we only consider the intrinsic geometry. **(3)** We explained in the paper how the hyperbolic and spherical cases are special cases of $\mathcal{Q}_\beta^{p,q}$ (lines 87-88 and lines 68-70). We will make it more explicit as suggested. **(4)** Our code is in the supp. material and will be publicly available. We reported some training times in the supp. material (line 540). On Zachary's dataset, the (Euclidean) optimizer in Section 4.1 is $10\%$ faster than the optimizer in Section 4.2 (165 vs 182 seconds) in the setup of line 540 since it requires less computations. We will report the comparisons. **(5)** Lastly, we will explicitly mention that $\forall \mathbf{x} \in \mathcal{Q}_\beta^{p,q}$, $g_\mathbf{x}(\cdot,\cdot) = \langle\cdot,\cdot\rangle_q$ where $g_\mathbf{x} : T_\mathbf{x}\mathcal{Q}_\beta^{p,q} \times T_\mathbf{x}\mathcal{Q}_\beta^{p,q} \to \mathbb{R}$.

**R3**: **(1)** Thank you for mentioning Feragen's work that was among the first to study tree-data in the CV and ML community. We will cite it in the introduction when we mention other machine learning works that were also heavily inspired by Gromov's work. Nonetheless, Feragen et al. consider CAT(0) spaces (e.g. hyperbolic spaces). Our work generalizes both hyperbolic and spherical spaces, the latter is not CAT(0). (**R4**,**R3**) **(2) Motivation of Eq. (9):** As explained in the paper, there exist pairs of points $\mathbf{x}, \mathbf{y} \in \mathcal{Q}_\beta^{p,q}$ for which $\log_\mathbf{x}(\mathbf{y})$ is not defined. Eq. (9) approximates the dissimilarity when $\log_\mathbf{x}(\mathbf{y})$ is not defined but other choices are possible. When a geodesic does not exist, a standard way in differential geometry to calculate curves (and distances) is to consider broken geodesics. One might then consider instead the dissimilarity $\mathsf{d}_\gamma(\mathbf{x}, -\mathbf{x}) + \mathsf{d}_\gamma(-\mathbf{x}, \mathbf{y}) = \pi\sqrt{|\beta|} + \mathsf{d}_\gamma(-\mathbf{x}, \mathbf{y})$ if $\log_\mathbf{x}(\mathbf{y})$ is not defined (see line 422 of the supp. material) since $-\mathbf{x} \in \mathcal{Q}_\beta^{p,q}$ and $\log_{-\mathbf{x}}(\mathbf{y})$ is defined. **(3)** We disagree about the fact that we used hacks to create symmetric weights. Our second dataset has an undirected (hence symmetric) weight matrix by default. Moreover, in Zachary's paper, $\mathbf{C}$ was constructed in an *ad hoc* manner and is almost identical to its transpose (i.e. almost symmetric). The weight matrix $\mathbf{C}$ is illustrated in Fig. 3 of Zachary's paper. Our symmetrized matrix $\mathbf{S} = \mathbf{C} + \mathbf{C}^\top$ is very similar to $2\mathbf{C}$, which is why we considered it. In conclusion, our approach can be applied to any undirected weighted graph.

(**R2**,**R3**,**R4**) **Motivation of ultrahyperbolic representations for graphs:** The choice of geometry to represent graphs is still an open problem in general. It depends on the topology of the graph and the kind of relationships between nodes. For instance, hyperbolic geometry was mathematically shown to be appropriate for tree-like graphs, but not for other types of graphs. Ultrahyperbolic geometry has the advantage of generalizing both hyperbolic and spherical geometries and can describe relationships specific to those geometries. In particular, the geodesic *distance* can be written in the same way as the Poincaré and spherical distances as shown in Eq. (8); some parts of the manifold are hyperbolic or spherical as explained in the paper. The converse is not true. The framework might then automatically learn representations to be part of a same hyperbolic or spherical part of the manifold depending on the context. Those reasons led us to consider hierarchical graphs that were similar to trees, but where the presence of cycles in the graph limited the relevance of hyperbolic geometry. We experimentally validated our intuition. The choice of geometry with constant nonzero curvature (i.e. the optimal number of time and space dimensions $q$ and $p$) then seems to depend on how much the graph is similar to a tree or to a graph where spherical geometry is appropriate, such as cycle graphs (or a mix of both). We would also like to emphasize that ultrahyperbolic geometry can describe some graph concepts differently, if not better, than hyperbolic and spherical geometries. For instance, it is known that *triadic closure* is a concept in social network theory that is too extreme to hold true across very large, complex networks. In other words, if $(\mathbf{x}, \mathbf{y})$ and $(\mathbf{x}, \mathbf{z})$ are strongly tied, triadic closure would induce that $(\mathbf{y}, \mathbf{z})$ are strongly tied. As explained in line 144, the fact that we can find triplets that satisfy $\mathsf{d}_\gamma(\mathbf{x}, \mathbf{y}) = \mathsf{d}_\gamma(\mathbf{x}, \mathbf{z}) = 0$ but $\mathsf{d}_\gamma(\mathbf{y}, \mathbf{z}) > 0$ avoids triadic closure.

[Meta-Review · NeurIPS 2020]

The paper was carefully reviewed by four domain experts, who found in particular the theoretical contributions sound and useful. Both the reviews and the post-rebuttal discussion indicated, however, that the authors would very much have liked to see some indication, intuition or understanding of for which type of this particular representation would be of use. I encourage the authors to think about this for the camera ready version; this could significantly improve the impact of the paper.